# Null-Space Embedding for Driftless Invisible Image Watermarking

## Abstract

Recent progress in text-to-image diffusion highlights the need for invisible, tamper-resilient watermarking that maintains both visual fidelity and prompt alignment. Existing approaches often compromise on robustness, imperceptibility, or scalability, with many introducing semantic drift that weakens provenance guarantees. To address this, we introduce *NullGuard*, a training-free, plug-and-play watermarking framework that embeds cryptographically keyed signals in the null-space of pretrained diffusion Jacobians, using user-specific rotations to define imperceptible directions. A lightweight Gauss–Newton pivot refinement, constrained by a perceptual mask, perturbs only watermark-relevant components while preserving global semantics, and a likelihood-ratio test detects watermarks without DDIM inversion, achieving up to 99% detection accuracy under attacks such as cropping, blurring, and JPEG compression, with PSNR $\geq$ 45 dB. Extensive evaluations on MS-COCO and DiffusionDB demonstrate that NullGuard surpasses state-of-the-art (SOTA) methods in robustness, invisibility, and semantic alignment, offering a scalable foundation for provenance-aware diffusion governance. Anonymous Code: https://anonymous.4open.science/r/NullGuard-7766.

## Introduction

The rise of text-to-image diffusion models has transformed creative workflows by enabling users to generate photorealistic and semantically rich images directly from text Chang et al. (2023); Podell et al. (2023); Ramesh et al. (2022); Rombach et al. (2022); Saharia et al. (2022), fueling adoption across digital art Huang et al. (2022); Podell et al. (2023); Wang et al. (2025a), entertainment Blattmann et al. (2023a;b); Ho et al. (2022); Xing et al. (2024), advertising, and design. Yet, the same accessibility raises critical challenges around ownership, authenticity, and the potential for malicious use of synthetic content[1], prompting growing interest in provenance standards and traceability Ci et al. (2024); Huang et al. (2024); Zhang et al. (2024). Among the proposed combatant solutions, invisible watermarking has emerged as a central mechanism for copyright protection, authorship verification, and user-level traceability. Current methods fall into three types: post-processing Cox et al. (2008); Zhang et al. (2019), which alters pixels but sacrifices robustness; fine-tuning Cui et al. (2025); Fernandez et al. (2023); Liu et al. (2023); Xiong et al. (2023); Zhao et al. (2023b), which modifies model weights and limits scalability; and latent-space watermarking Wen et al. (2023), which preserves fidelity but may disrupt sampling.

Recent works such as Tree-Ring Wen et al. (2023), Stable Signature Fernandez et al. (2023), and Gaussian Shading Yang et al. (2024) attempt to balance semantic alignment and imperceptibility carefully. However, iterative denoising in many generative methods often introduces drift, which means the methods are unable to fully preserve the semantic alignment. This particular gap emphasizes the need for robust and semantically stable watermarking solutions.

In this work, we address the challenge of semantic drift in diffusion-based watermarking by introducing a latent-space embedding strategy that minimizes modification to the semantic content of the image. Moreover, rather than relying on full denoising simulation, NullGuard minimally perturbs the clean latent embedding along a key-modulated null-space direction, ensuring high semantic fidelity without compromising prompt alignment. NullGuard supports per-user cryptographic watermark

---

[1]https://edition.cnn.com/2023/05/22/tech/twitter-fake-image-pentagon-explosion/index.html

personalization via key-derived orthogonal rotations, enabling scalable deployment in real-world multi-user scenarios. To boost robustness, we perform multi-band frequency decomposition of the keyed direction—splitting it into coordinated high-, mid-, and low-frequency components to distribute energy across resilient spectral channels. In addition, a likelihood-ratio segmentation (LRS) masking stage selects low impact latent pixels for safe injection; we then apply a Gauss–Newton pivot refinement constrained by this LRS perceptual mask to restrict updates to watermark-relevant regions, thereby preserving the original image content. Our approach ensures the resulting watermarked images are virtually indistinguishable from original images, both visually and semantically, effectively mitigating the drift observed in prior methods. In summary, our main contributions are as follows:

- We introduce *NullGuard*, A watermarking framework that embeds imperceptible signals in the *spectral null-space* of pre-trained diffusion Jacobians. A secret key produces an orthogonal rotation that personalizes the null-space basis, enabling scalable per-user provenance without fine-tuning or inference overhead.

- A lightweight, closed-form Gauss–Newton pivot constrained by a likelihood-ratio segmentation mask limits edits to watermark relevant latent components, keeping PSNR $\geq 45$ dB while minimizing visual drift.

- We derive a statistically grounded likelihood-ratio test on the forward diffusion ELBO that detects watermarks with 99% watermark detection rate under cropping, JPEG compression, and noise, surpassing SOTA methods.

## RELATED WORK

### DIFFUSION MODELS AND LATENT INVERSION

Diffusion models have achieved remarkable success in high-fidelity image generation and now serve as a cornerstone in generative tasks such as text-to-image synthesis Avrahami et al. (2023); Nichol et al. (2021); Podell et al. (2023); Rombach et al. (2022), image editing Chen et al. (2024); Cho et al. (2024); Hertz et al. (2022); Mokady et al. (2023), and image restoration Xia et al. (2023); Yue et al. (2024); Islam et al. (2024); Jiang et al. (2023). A prominent framework is the denoising diffusion probabilistic model (DDPM) Ho et al. (2020), which generates data by reversing a Markovian noising process where Gaussian noise is added progressively over $T$ steps based on a fixed scheduling. Latent diffusion models (LDMs) Rombach et al. (2022) improve computational efficiency by performing the generative process in a compressed latent space, where an encoder $\mathcal{E}$ maps an image to a latent vector and a decoder $\mathcal{D}$ reconstructs the image. The generation proceeds by denoising a Gaussian prior latent $z_T$ toward $z_0$, which is then decoded into the image domain. DDIMs enable deterministic, non-Markovian sampling and inversion, allowing recovery of $z_0$ from generated images, which is critical for consistent watermark verification.

### IMAGE WATERMARKING IN DIFFUSION MODELS

In generative diffusion models, watermarking techniques fall into two main categories: post-processing and in-generation methods. Post-processing approaches embed the watermark after image synthesis using frequency domain transforms like DWT Xia et al. (1998), DCT Miller et al. (2008), or encoder-based schemes such as HiDDeN Zhu et al. (2018) and StegaStamp Tancik et al. (2020), but often degrade visual quality and remain vulnerable to perturbations. In-generation methods embed watermarks during generation by modifying model internals or latent variables, as seen in the Stable Signature Fernandez et al. (2023), which fine-tunes the decoder for watermark extraction but limits scalability because each extra key bit demands an additional output neuron; thus, larger keyspaces require proportionally larger (re-trained) decoders. Lightweight techniques like Tree-Ring Wen et al. (2023) avoid retraining by modifying latent frequencies, though this can distort semantics and reduce generation diversity. Recent works such as Gaussian Shading Yang et al. (2024) and ZoDiac Zhang et al. (2024) improve fidelity and robustness, but still struggle with geometric distortions, control limitations, or high computational overhead.

Figure 1: **Overview of our NullGuard Architecture.** For watermark fabrication, the null space $W$ and user-exclusive rotation matrix provides the bedrock of the watermark signature followed by its refinement (frequency re-weighting, likelihood-ratio segmentation, and semantics-preserving latent optimization). During verification, the image is analyzed by statistical testing, including the forward-score gap.

## PROPOSED METHOD: NULLGUARD

We introduce *NullGuard*, an invisible watermarking method that embeds robust signals using null-space projection and secret-key rotation. By minimally altering the diffusion trajectory, NullGuard preserves image semantics while ensuring watermark resilience. We begin with the problem formulation, then outline the method and its core components.

### PROBLEM FORMULATION

The task of image watermarking in generative models can be described as follows: given an image, the objective is to embed a hidden signal that is imperceptible to humans yet reliably detectable by an authorized verifier. The embedding process must preserve both perceptual fidelity and semantic integrity, ensuring that the image remains visually indistinguishable from its original form. Verification involves designing a reliable statistical test to determine whether a given image contains a watermark. This verification should be robust against common post-processing operations (e.g., compression, noise, cropping). Moreover, in practical scenarios, the system should support user-specific watermarking, enabling different users to be assigned distinct, verifiable signals derived from their unique keys. Two overarching constraints govern the formulation: *invisibility*, which ensures that watermarks do not alter visual or semantic quality, and *robustness*, which ensures that the watermark signal persists under a wide range of benign and adversarial transformations.

### PROPOSED FRAMEWORK

**Null-Space Generation.** Let $x$ be the clean RGB image to be watermarked. NullGuard operates directly in the latent diffusion's zero-noise space, also equivalently known as the VAE's latent space $\mathcal{Z}$ Kingma & Welling (2013). Given the latent encoding $z^*$ of $x$, offsetting it with an arbitrary watermark signal is prone to distort existing visual semantics. To both apply robust and semantics-preserving watermark signal, it is essential to identify a universal latent watermark basis that explicitly discards semantic axes and contains insensitive axes. Towards this goal, NullGuard employs a semantics null-space construction strategy leveraging multiple diffusion model backbones. Namely, we first aggregate the hyperfeatured (spanning across diffusion time and space) Jacobian matrices of mean output $\bar{\varepsilon}_{\theta(f)}$, each denoted as follows:

$$\mathbf{J}_t^{(f)} = \frac{\partial \bar{\varepsilon}_{\theta(f)}(z_t, e)}{\partial e^{[\text{EOS}]}} \in \mathbb{R}^{e_{dim}} \tag{1}$$

where $z_t$ denotes the noisy latents at timestep $t$, $e$ is the unconditional text embedding (or null-text embedding in the T2I context), and $e^{[\text{EOS}]}$ for the "EOS" token. By aggregating these Jacobian matrices across various timesteps and diffusion backbones, we construct a comprehensive Jacobian matrix $\mathcal{J}$ as follows:

$$\mathcal{J} = \left(\mathbf{J}_{t_1}^{(f_1)}, \ldots, \mathbf{J}_{t_n}^{(f_{k-1})}, \ldots, \mathbf{J}_{t_n}^{(f_k)}\right)^{\top} \in \mathbb{R}^{e_{dim} \times N}, \tag{4}$$

---

**Algorithm 1: Key → Rotation Matrix.**

---

**Require:** 256-bit key $K$, null-space dimension $m$
**Ensure:** Rotation matrix $Q_K \in \mathrm{SO}(m)$
  1: $s \leftarrow \texttt{SHAKE-256}(K)$             ▷ 32-byte cryptographic seed
  2: Initialize PRNG with seed $s$
  3: $A \leftarrow \mathrm{PRNG.rand}(m, m) \times 2 - 1$          ▷ entries in $(-1, 1)$
  4: $(Q, R) \leftarrow \mathrm{QR}(A)$               ▷ orthonormalize
  5: **if** $\det(Q) < 0$ **then**
  6:      $Q[:, 1] \leftarrow -Q[:, 1]$
  7: **end if**
  8: **return** $Q$

---

where $N$ represents the total number of sampled latent states across all models and timesteps. To identify the watermark embedding directions, we apply randomized Singular Value Decomposition (SVD) Golub & Van Loan (2013) to the aggregated $\mathcal{J}$, yielding the factorization $\mathcal{J} \approx U\Sigma V^{\top}$. From this decomposition, we select the right-singular vectors associated with near-zero singular values to construct a robust latent null-space as $W = \{v_{r+1}, v_{r+2}, \ldots, v_{r+m}\} \subseteq V$, in $\mathbb{R}^{e_{dim} \times m}$ with $m$ representing the chosen dimension of the null-space, and $r$ the rank of $\mathcal{J}$. Notably, the semantics-orthogonality of this space spans macroscopically over the whole diffusion process. Once in To ensure cryptographic robustness, the null-space basis $W$ is subsequently rotated by a secret orthogonal matrix $Q_K$, derived deterministically from a user-specific cryptographic key $K$ given by:

$$W_K = WQ_K, \quad Q_K \in \mathrm{SO}(m). \tag{2}$$

This secret-rotated basis $W_K$ uniquely defines the watermark embedding directions for each user, offering cryptographic security and $1^{st}$-order imperceptibility. The constructed null-space thus serves as a stable and secure foundation for embedding robust, invisible watermarks, significantly facilitating downstream semantic optimization and reliable verification under diverse adversarial manipulations.

**Key-to-Secret Rotation.** To convert a user key $K \in \{0, 1\}^{256}$ into an orthogonal matrix $Q_K$ we adopt a deterministic *hash → seed → QR* pipeline as detailed in Algorithm 1, where lines 1-3 expand the key $K$ into a high-entropy matrix; the QR decomposition Golub & Van Loan (2013) guarantees orthogonality, and the determinant flip (line 6) ensures membership in $\mathrm{SO}(m)$. Because this mapping is stateless and deterministic, the encoder and verifier need only share the key $K$, never the matrix itself. Applying $Q_K$ to a fixed null-space basis $W \in \mathbb{R}^{M \times m}$ yields the keyed basis $W_K = WQ_K$; its first column $w_K$ is the watermark direction subsequently shaped spectrally (multi-band) and spatially (LRS masked) before insertion.

**Multi-Band Frequency Shaping of the Watermark Bump.** NullGuard distributes watermark energy across three complementary spectral bands—high $w_K^{\mathrm{HF}}$, mid $w_K^{\mathrm{MF}}$, and low $w_K^{\mathrm{LF}}$—as in Eq. 3, to enhance robustness under heterogeneous manipulations while remaining imperceptible. We project the keyed vector $w_K$ to the Fourier domain and apply a circular high-pass stopband (a disk of radius $\rho_{\mathrm{HP}}$ around the DC origin) to obtain $w_K^{\mathrm{HF}}$. The resulting component, unit-$\ell_2$ normalised, contributes high-detail structure that aids robustness to local crops and mild rescaling. For mid-frequencies, we use an analytic cosine field $\cos\left(\frac{\pi}{2}y\right)\cos\left(\frac{\pi}{2}x\right), \quad (x, y) \in [-1, 1]^2$, defined on the latent grid (replicated across channels) and then unit-normalised, with strength controlled by $\beta_{\mathrm{MF}}$. The low-frequency term $w_K^{\mathrm{LF}}$ is a (quasi-)DC component (constant field prior to masking), unit-energy and scaled by $\beta_{\mathrm{LF}}$. The three unit-$\ell_2$ components are blended *before* masking as follows, which ensures a nonnegative HF weight.

$$\tilde{w} = \left(1 - \beta_{\mathrm{MF}} - \beta_{\mathrm{LF}}\right)w_K^{\mathrm{HF}} + \beta_{\mathrm{MF}}w_K^{\mathrm{MF}} + \beta_{\mathrm{LF}}w_K^{\mathrm{LF}}, \quad 0 \le \beta_{\mathrm{MF}}, \beta_{\mathrm{LF}}, \; \beta_{\mathrm{MF}} + \beta_{\mathrm{LF}} \le 1, \tag{3}$$

Before insertion, the blended template $\tilde{w}$ is spatially gated by the likelihood-ratio segmentation mask $M$ (selecting low-impact latent pixels) and then rescaled so that $\|\Delta z\|_2 = |\alpha|$, where $\Delta z = \alpha\left(M \odot \tilde{w}\right)$. An entropy-aware line search (bisection in practice) selects the largest $|\alpha|$ that satisfies the perceptual budget $\mathrm{LPIPS}(x, \hat{x}) \le \texttt{TARGET\_LPIPS}$ (0.008). Finally, a *mask-aware* Gauss–Newton pivot refines the scalar payload $\alpha$ while freezing masked watermark pixels, preserving semantics and improving downstream detectability.

---

**Algorithm 2: NullGuard Watermark Embedding**

---

**Require:** clean latent $z^\star$, key $K$, initial scalar $\alpha$, LRS mask $M$, iterations $N$, regulariser $\eta$
**Ensure:** refined latent $z^{\text{wm}}$ and final $\alpha$

    *// keyed direction in the null–space*
1: $w_K \leftarrow \textbf{first\_col}(W\, Q_K)$                   $\triangleright$ $W$ pre-computed basis; $Q_K$ from key
    *// precompute band components (unit-$\ell_2$) once*
2: $\left(w_K^{\text{HF}}, w_K^{\text{MF}}, w_K^{\text{LF}}\right) \leftarrow \textbf{BandDecompose}(w_K)$   $\triangleright$ HF high-pass; MF analytic cosine; LF quasi-DC; each unit norm
    *// helper to rebuild the blended-and-masked bump for any $\alpha$*
3: **function** BUMP($\alpha$)
4:     $\tilde{w} \leftarrow \left(1 - \beta_{\text{MF}} - \beta_{\text{LF}}\right) w_K^{\text{HF}} + \beta_{\text{MF}}\, w_K^{\text{MF}} + \beta_{\text{LF}}\, w_K^{\text{LF}}$
5:     **return** $\alpha\, (M \odot \tilde{w})$
6: **end function**
7: $\Delta z \leftarrow$ BUMP($\alpha$)
8: $W \leftarrow \text{diag}(1 - M)$               $\triangleright$ zero weight on watermark pixels (mask complement)
9: $J \leftarrow w_K[:C] \otimes \mathbf{1}_{hw}$            $\triangleright$ replicate channel-wise along spatial grid
10: **for** $k = 1$ **to** $N$ **do**
11:     $r \leftarrow \text{vec}(\Delta z)$
12:     $\delta\alpha \leftarrow -\dfrac{J^\top W\, r}{J^\top W\, J + \eta}$
13:     $\alpha \leftarrow \alpha + \delta\alpha$
14:     $\Delta z \leftarrow$ BUMP($\alpha$)
15: **end for**
16: $z^{\text{wm}} \leftarrow z^\star + \Delta z$
17: **return** $z^{\text{wm}}$

---

LIKELIHOOD-RATIO SEGMENTATION (LRS). Before the Gauss–Newton pivot refinement, we compute a binary mask $M \in \{0, 1\}^{C \times h \times w}$ that pinpoints latent pixels able to host watermark energy with minimal perceptual risk. Let $d_i = z_i^{\text{wm}} - z_i^\star$ denote the initial latent difference at latent pixel $i$. Under the null hypothesis $H_0 : d_i \sim \mathcal{N}(0, \sigma^2)$ and an alternative mean shift $\Delta_i$, the per-latent pixel log-likelihood ratio is as follows:

$$\Lambda_i = \frac{d_i\, \Delta_i}{\sigma^2} - \frac{\Delta_i^2}{2\sigma^2}. \tag{4}$$

We transform each latent pixel-wise statistic $\Lambda_i$ into a two-sided $p$-value via $p_i = 2\, \Phi(-|\Lambda_i|)$, where $\Phi$ denotes the standard normal cumulative distribution function. To control the false discovery rate (FDR), we apply the Benjamini–Hochberg (BH)[2] procedure at level $q$. Letting $p_{(1)} \leq \ldots \leq p_{(m)}$ denote the sorted $p$-values across all $m = C \cdot h \cdot w$ latent pixels, we identify the largest index $j^*$ such that $p_{(j^*)} \leq (j^* \cdot q)/m$, and set the selection threshold $\tau = p_{(j^*)}$. The final binary mask is then defined by $M_i = \mathbf{1}[p_i \leq \tau]$, ensuring that the false-positive rate $\Pr[M_i = 1 \wedge H_0]$ remains bounded by $q$. This statistically grounded mask is then passed to the Gauss–Newton pivot refinement stage, guiding watermark edits to latent pixels that genuinely support the watermark signal while preserving perceptual fidelity.

**Semantics-Preserving Latent Optimisation.** Embedding a watermark in latent space is useful only if the resulting image remains visually indistinguishable from the original. Therefore, Null-Guard runs a compact optimization that adjusts only a single scalar payload $\alpha$, leaving every other network weight and latent coefficient fixed. The procedure is inspired by gradient-guided latent editing in diffusion models, yet differs in two key respects. First, the watermark bump is already localized by an LRS mask $\mathcal{M} \in \{0, 1\}^{C \times h \times w}$ computed once with the initial latent difference; latent pixels marked by the mask $\mathcal{M}$ are irrevocably reserved for the watermark. Second, because the payload $\alpha$ is a scalar, the full non-linear optimisation collapses to a 1-D *Gauss–Newton pivot (GNP)* Nocedal & Wright (2006) that can be written in closed form and evaluated in milliseconds. Two complementary objectives shape this update.

*(i) Semantic Consistency.* The latent optimization stage explicitly preserves semantic fidelity by minimizing any unintended drift between the watermarked latent trajectory and its clean counterpart. Specifically, in Algorithm 2, lines 8-11, a diagonal weighting matrix $W = \text{diag}(1 - \mathcal{M})$ (line 8) is

---

[2]https://en.wikipedia.org/wiki/False_discovery_rate#BH_procedure

Table 1: **Image quality and watermark detection performance of NullGuard versus SOTA methods**, where the best value from SOTA is in gray and ours in green . We report average PSNR, SSIM, and LPIPS for the watermarked images before any attack (*Clean*), and the WDR under a range of common post-attack distortions: brightness change, contrast change, JPEG compression, additive Gaussian noise, Gaussian blur, BM3D filtering, two state-of-the-art neural compression attacks (Bmshj18, Cheng20), a diffusion-space inversion attack (Zhao23), the combined "All w/o ↻" setting, and pure 90° rotation.

| Method | Image Quality | | | Clean | WDR (Post-Attack) | | | | | | | | | | |
|---|---|---|---|---|---|---|---|---|---|---|---|---|---|---|---|
| | PSNR↑ | SSIM↑ | LPIPS↓ | | Bright | Contrast | JPEG | G-Noise | G-Blur | BM3D | Bmshj18 | Cheng20 | Zhao23 | All w/o ↻ | Rot.(↻) |
| **MS-COCO** | | | | | | | | | | | | | | | |
| DwtDct | 37.88 | 0.97 | 0.02 | 0.790 | 0.000 | 0.000 | 0.000 | 0.687 | 0.156 | 0.000 | 0.000 | 0.000 | 0.000 | 0.000 | 0.000 |
| DwtDctSvd | 38.06 | 0.98 | 0.02 | 1.000 | 0.098 | 0.100 | 0.746 | 0.998 | 1.000 | 0.452 | 0.016 | 0.032 | 0.124 | 0.000 | 0.000 |
| RivaGAN | 40.57 | 0.98 | 0.04 | 1.000 | 0.996 | 0.984 | 1.000 | 1.000 | 1.000 | 0.974 | 0.010 | 0.010 | 0.032 | 0.000 | 0.000 |
| SSL | 41.81 | 0.98 | 0.06 | 1.000 | 0.992 | 0.996 | 0.046 | 0.038 | 1.000 | 0.000 | 0.000 | 0.000 | 0.000 | 0.000 | 0.952 |
| CIN | 41.77 | 0.98 | 0.02 | 1.000 | 1.000 | 1.000 | 0.944 | 1.000 | 1.000 | 0.580 | 0.662 | 0.666 | 0.478 | 0.000 | 0.216 |
| StegaStamp | 28.64 | 0.91 | 0.13 | 1.000 | 0.998 | 0.998 | 1.000 | 0.998 | 1.000 | 0.998 | 0.998 | 1.000 | 0.286 | 0.002 | 0.000 |
| ZoDiac | 29.41 | 0.92 | 0.09 | 0.998 | 0.998 | 0.998 | 0.992 | 0.996 | 0.996 | 0.994 | 0.992 | 0.986 | 0.988 | 0.510 | 0.538 |
| **NullGuard** | 53.95 | 0.999 | 0.008 | 0.999 | 1.000 | 1.000 | 0.998 | 0.996 | 0.998 | 0.998 | 0.998 | 0.996 | 0.990 | 0.732 | 0.740 |
| **DiffusionDB** | | | | | | | | | | | | | | | |
| DwtDct | 37.77 | 0.96 | 0.02 | 0.690 | 0.000 | 0.000 | 0.000 | 0.574 | 0.224 | 0.000 | 0.000 | 0.000 | 0.000 | 0.000 | 0.000 |
| DwtDctSvd | 37.84 | 0.97 | 0.02 | 0.998 | 0.088 | 0.088 | 0.812 | 0.982 | 0.996 | 0.686 | 0.014 | 0.030 | 0.116 | 0.000 | 0.000 |
| RivaGAN | 40.60 | 0.98 | 0.04 | 0.974 | 0.932 | 0.932 | 0.898 | 0.958 | 0.966 | 0.858 | 0.008 | 0.004 | 0.024 | 0.000 | 0.000 |
| SSL | 41.84 | 0.98 | 0.06 | 0.998 | 0.990 | 0.996 | 0.040 | 0.030 | 1.000 | 0.000 | 0.000 | 0.000 | 0.000 | 0.000 | 0.898 |
| CIN | 39.99 | 0.98 | 0.02 | 1.000 | 1.000 | 1.000 | 0.942 | 0.998 | 0.998 | 0.624 | 0.662 | 0.660 | 0.498 | 0.002 | 0.212 |
| StegaStamp | 28.51 | 0.90 | 0.13 | 1.000 | 0.998 | 0.998 | 1.000 | 0.998 | 0.998 | 1.000 | 0.996 | 0.998 | 0.302 | 0.002 | 0.000 |
| ZoDiac | 29.18 | 0.92 | 0.07 | 1.000 | 0.998 | 0.998 | 0.994 | 0.998 | 1.000 | 1.000 | 0.994 | 0.992 | 0.988 | 0.548 | 0.558 |
| **NullGuard** | 54.10 | 0.999 | 0.008 | 1.000 | 1.000 | 1.000 | 0.998 | 0.994 | 0.998 | 1.000 | 1.000 | 0.996 | 0.994 | 0.778 | 0.781 |

Table 2: Effect of the lock-in denoising step count on image quality (PSNR↑) and WDR↑ of our NullGuard on the MS-COCO split. "Rotation" and "Zhao23" are two of the hardest attacks; "All w/o ↻" is the average WDR over all non-rotational attacks.

| Denoising Steps ($T$) | PSNR↑ | WDR↑ before and after attack | | | |
|---|---|---|---|---|---|
| | | Pre Attack | Post-Attacks | | |
| | | | Rotation | Zhao23 | All w/o ↻ |
| 50 | 28.1 | 0.999 | 0.80 | 0.997 | 0.78 |
| 10 | 42.8 | 0.999 | 0.78 | 0.995 | 0.76 |
| 1 (**default**) | 53.9 | 0.999 | 0.74 | 0.990 | 0.73 |
| 0 | 54.4 | 0.975 | 0.32 | 0.820 | 0.05 |

constructed from the LRS mask $\mathcal{M}$. This mask, which is derived statistically, precisely identifies the latent pixels that encode the watermark information. Consequently, its complement $(1 - \mathcal{M})$ identifies latent pixels that exclusively represent semantic content. By weighting the latent difference $r = z^{\mathrm{wm}} - z^{\star}$ (line 11) through this mask complement, the optimization residual is restricted to non-watermark regions only. Mathematically, minimizing the quadratic form $r^{\top} W r$ thus becomes equivalent to explicitly minimizing the masked $\ell_2$ semantic drift as follows:

$$\|(1 - \mathcal{M}) \odot (z^{\star} - z^{\mathrm{wm}})\|_2^2. \tag{5}$$

Through this mechanism, every incremental adjustment to the scalar watermark embedding parameter $\alpha$ (as computed by the subsequent GNP update) reduces semantic perturbation strictly within regions free from watermark-related modifications. Consequently, the final watermarked image maintains near-perfect alignment with the semantic content of the original input image, preserving its perceptual and semantic integrity. Proof is in Supplementary Material.

*(ii) Watermark Preservation.* While semantic fidelity is prioritized in non-watermark latent pixels, robust watermark preservation is enforced explicitly within watermark latent pixels indicated by the LRS mask $\mathcal{M}$. To accomplish this, the same diagonal weighting matrix $W = \mathrm{diag}(1 - \mathcal{M})$ completely eliminates watermark latent pixels from influencing the optimization update. Specifically, during the closed-form scalar update step (line 12) of the algorithm, the scalar GNP solution is computed $\delta\alpha^{\star}$. In line 12, $J$ is the flattened keyed-watermark direction, $r$ is the current latent residual, and $\eta$ is a numerical regularization parameter.

By construction, any latent pixel flagged as a watermark latent pixel ($\mathcal{M}_i = 1$) contributes zero rows and columns to the weighting matrix $W$. Consequently, watermark latent pixels do not influence the scalar optimization step, effectively locking their values. This explicit masking ensures that the condition $\mathcal{M} \odot (z^{\mathrm{wm}} - \Delta z) \approx \mathbf{0}$ remains exactly satisfied throughout the optimization (lines 12-14).

Table 3: Influence of the decision threshold ($p^*$) on our detection performance. $p^* = 0.9$ is used in our default settings.

| $p^*$ | FPR↓ | WDR↑ before / after attack | | | |
|---|---|---|---|---|---|
| | | Clean | Rotation | Zhao23 | All w/o ↺ |
| 0.90 | 0.060 | **0.999** | **0.740** | **0.990** | **0.732** |
| 0.95 | 0.028 | 0.998 | 0.604 | 0.989 | 0.590 |
| 0.99 | **0.005** | 0.994 | 0.320 | 0.955 | 0.301 |

Table 4: PSNR of attacked watermarked images and WDR of ZoDiac and StegaStamp under varying **Brightness** (left) and **Contrast** (right) strengths from 0.2 to 1.0 (no attack).

| | Detection Threshold | FPR↓ | WDR under Brightness Factor of | | | | | WDR under Contrast Factor of | | | | |
|---|---|---|---|---|---|---|---|---|---|---|---|---|
| | | | 0.2 | 0.4 | 0.6 | 0.8 | 1.0 | 0.2 | 0.4 | 0.6 | 0.8 | 1.0 |
| **PSNR** | – | – | 11.41 | 14.56 | 27.95 | 33.58 | 100.0 | 14.54 | 23.03 | 30.56 | 36.57 | 100.0 |
| StegaStamp | (61/96) | 0.056 | 0.852 | 0.988 | 1.0 | 1.0 | 1.0 | 0.730 | 0.980 | 1.0 | 1.0 | 1.0 |
| StegaStamp | (60/96) | 0.094 | 0.890 | 1.0 | 1.0 | 1.0 | 1.0 | 0.768 | 0.998 | 1.0 | 1.0 | 1.0 |
| ZoDiac | $p^* = 0.95$ | 0.032 | 0.994 | 0.994 | 0.998 | 0.998 | 0.998 | 0.990 | 0.996 | 0.998 | 0.998 | 0.998 |
| ZoDiac | $p^* = 0.90$ | 0.062 | 0.996 | 0.998 | 0.998 | 0.998 | 0.998 | 0.994 | 0.998 | 0.998 | 0.998 | 0.998 |
| **NullGuard** | $p^* = 0.90$ | 0.060 | 1.000 | 1.000 | 1.000 | 1.000 | 1.000 | 1.000 | 1.000 | 1.000 | 1.000 | 1.000 |

Thus, watermark latent pixels remain strictly untouched during latent adjustments, firmly embedding watermark information without compromising its detectability.

By combining semantic preservation outside the mask and strict watermark protection within, the optimization stage of NullGuard achieves an optimal balance, minimizing semantic drift and simultaneously ensuring watermark robustness. After the latent optimization described above, our NullGuard performs a final *zero-noise* denoising step, which is often referred to as a "lock-in" operation, to ensure that the optimized watermark perturbation is correctly and robustly propagated into the final latent representation $\hat{z}_0$. During optimization, updates are restricted exclusively to the non-watermark latent pixels identified by the mask complement $(1 - \mathcal{M})$, whereas the latent pixels selected by the mask itself (where $\mathcal{M} = 1$) remain unchanged, thereby strictly preserving the embedded watermark signal.

**Verification (Forward Score Gap).** Let $z^{wm} \in \mathbb{R}^{C \times h \times w}$ be the (already watermarked) latent obtained by VAE-encoding the suspect image, and let $e_0$ denote the text-conditioning embedding of the null-prompt. For a user key $K$, we construct a keyed direction $w_K = \text{first\_col}(W_K)$ as in Sec.Key-to-Secret Rotation, which we use to offset $e_0$ to $e_K = e_0 + \gamma \cdot w_K$ with $\gamma > 0$. We score $(z_t, e_0)$ by running the diffusion model forward for $T$ scheduler steps and accumulating a per-pixel "forward score" $\varepsilon_\theta(z_t, e_0)$ with DDIM-inverted $z_t$ (an ELBO-like surrogate; for DDPM it is a standard ELBO term, for DDIM it is a deterministic proxy[3]). Our basic statistic is the keyed-text *forward score gap* as follows:

$$\mathcal{G} = \sum_t^T \frac{\varepsilon_\theta(z_t, e_K) - \varepsilon_\theta(z_t, e_0)}{C\,H\,W}, \qquad \text{expected } \mathcal{G} > 0 \text{ for the correct key } K. \tag{6}$$

We calibrate the noise floor once per deployment by drawing random keys and computing $\mathcal{G}$ with the same $(T, \gamma, \text{prompt})$, then set $\sigma_0 = \text{Std}[\mathcal{G}(K)]$, where $K \sim \text{Unif}(\{0,1\}^{256})$. At test time we apply a one-sided z-test with $S = \mathcal{G}/\sigma_0$ where and $p = 1 - \Phi(S)$; we accept iff $S > 0$ and $p \le \alpha$ (typically $\alpha \in \{0.05, 0.01\}$). This touches only the text embedding—no latent re-embedding—so verification is fast and non-invasive. More details is in Appendix.

# EXPERIMENTS

## EXPERIMENTAL SETUP

**Datasets.** For each domain, we randomly sample 500 images from widely-used benchmarks: **MS-COCO** Lin et al. (2014), and **DiffusionDB** Wang et al. (2022).

---

[3]With deterministic schedulers the transition variance is fixed; we therefore use a contrastive forward score whose constant offsets cancel in differences.

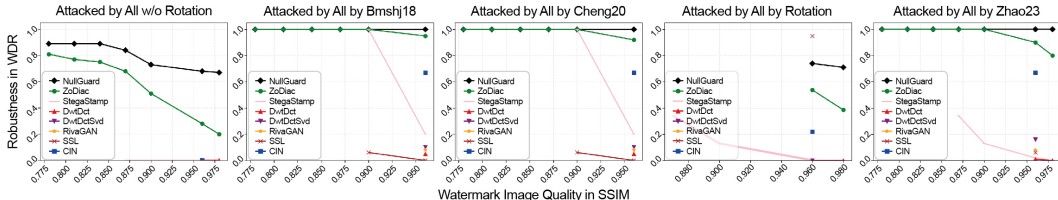

Figure 2: Trade-off between the watermarked image quality (SSIM) and the WDR on the MS-COCO dataset. The image quality is controlled by thresholds $s^* \in [0.8, 0.98]$, and the robustness is evaluated post-attack with four advanced attack methods.

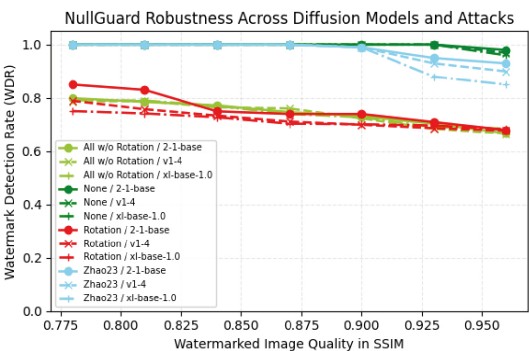

Figure 3: NullGuard exhibits comparable results when using different pre-trained stable diffusion models. Colors represent attacks; "None" represents no attack.

**NullGuard Configuration.** All experiments use the `stable-diffusion-2-1-base` checkpoint in half-precision on a single NVIDIA GPU, with 50 DDPM denoising steps unless `keep_pixels`=true, in which case a single zero-noise "lock-in" step is used. A universal 8D null-space basis (`NULLSPACE_DIM=8`) is precomputed and rotated using a 256-bit user key into a private basis $W_K = WQ_K$, where $Q_K \in \mathrm{SO}(8)$ is generated via Algorithm 1. The watermark scalar starts at $\sigma = 3.2 \times 10^{-3}$ and is adaptively reduced via bisection under a perceptual LPIPS budget of 0.008, with a hard cap at $8.0 \times 10^{-4}$. A multi-band latent bump with weights $0.4 : 0.4$ ($\beta_{\mathrm{MF}} : \beta_{\mathrm{LF}}$) is sparsified by a likelihood-ratio segmentation (LRS) mask and tiled with `TILE_SIZE=2` for redundancy. During refinement, 20 Gauss–Newton Pivot iterations are applied using $\eta = 5 \times 10^{-4}$, while verification is performed via a 20-step ELBO-gap test calibrated with dataset-specific noise $\sigma_0$.

**Watermarking Baselines.** We benchmark NullGuard against six prior watermarking approaches. Two frequency-domain classical methods, **DwtDct** and **DwtDctSvd** Shih (2017), are included. We also evaluate four learned watermarking models: **RivaGAN** Zhang et al. (2019), **SSL** Fernandez et al. (2022), **CIN** Ma et al. (2022), **StegaStamp** Tancik et al. (2020) and **ZoDiac** Zhang et al. (2024). These baselines offer a range of techniques, including adversarial training, invertible transformations, and attention mechanisms. Following prior protocols, we use 32-bit messages for Dwt-Dct, DwtDctSvd, RivaGAN, SSL, and CIN, and 96-bit messages for StegaStamp. Detection thresholds are set at $p < 0.01$, requiring correct detection on 24/32 or 61/96 bits, respectively Zhao et al. (2023a) in StegaStamp.

**Adversarial Transformations.** To assess robustness, we evaluate each method under a diverse set of adversarial image manipulations commonly used in the literature Zhao et al. (2023a); An et al. (2024). These transformations include brightness and contrast adjustments (scaling factor = 0.5), JPEG compression (quality = 50), 90-degree image rotation, additive Gaussian noise with standard deviation $\sigma = 0.05$, Gaussian blur with kernel size 5 and $\sigma = 1$, BM3D denoising with $\sigma = 0.1$, VAE-based compression Ballé et al. (2018); Cheng et al. (2020), and stable diffusion-based image regeneration Zhao et al. (2023b).

**Evaluation Metrics.** We report image quality using Peak Signal-to-Noise Ratio (PSNR), Structural Similarity Index (SSIM) Wang et al. (2004), and LPIPS Zhang et al. (2018), where higher PSNR/SSIM and lower LPIPS indicate better perceptual fidelity. For robustness, we use Watermark Detection Rate (WDR), equivalent to True Positive Rate.

EXPERIMENTAL RESULTS

IMAGE QUALITY AND ROBUSTNESS

Table 1 shows that NullGuard consistently achieves the best perceptual quality across MS-COCO and DiffusionDB, with PSNR $\geq$ 54 dB, SSIM $\approx$ 0.999, and LPIPS $\leq$ 0.008. On MS-COCO, it outperforms CIN by over 12 dB in PSNR while preserving semantic fidelity. Unlike StegaStamp and ZoDiac, which introduce visible artifacts, NullGuard's watermarks remain visually and perceptually imperceptible. NullGuard also demonstrates strong robustness under diverse image attacks. It maintains WDR $\geq$ 0.998 under JPEG compression and Gaussian blur, and 0.990 under the strong Zhao23 regeneration attack. It further sustains high WDRs under combined distortions and rotation, outperforming all baselines. As shown in Table 2, this robustness arises from a single-step lock-in refinement that efficiently propagates watermark signals while minimizing image alteration. Removing this step reduces WDR significantly, while increasing it harms visual quality. Together, these results highlight NullGuard's ability to balance fidelity and resilience without retraining or model modification.

ABLATION STUDIES

**Varying the SSIM Threshold** $s^*$. Figure 2 illustrates the robustness–quality trade-off for various watermarking methods under four strong attacks, as the SSIM threshold $s^*$ varies from 0.80 to 0.98. NullGuard consistently outperforms all baselines by maintaining high WDR across all attack scenarios, especially under compression and geometric distortions, while other methods like StegaStamp and DwtDct degrade significantly as image quality increases.

**Varying the Detection Threshold** $p^*$. Table 3 analyzes the effect of varying the detection threshold $p^*$ on NullGuard's watermark detection performance. Lowering $p^*$ increases robustness under attacks like Rotation and Zhao23, but comes at the cost of higher false positive rates (FPR), highlighting a trade-off between sensitivity and precision.

**Varying the Backbone Models.** Figure 3 shows that NullGuard maintains strong WDR across three diffusion backbones (2-1-base, v1-4, xl-base-1.0) under varying SSIM thresholds and four attack conditions. While WDR slightly declines under severe attacks like Zhao23 and Rotation, NullGuard remains consistently robust across all models, with 2-1-base showing the highest resilience.

**Varying the Attack Strength.** Table 4 compares the robustness of NullGuard, ZoDiac, and StegaStamp under varying brightness and contrast distortions from 0.2 to 1.0. While all methods achieve perfect WDR in the no-attack setting, NullGuard uniquely maintains a WDR of 1.000 across all attack strengths. ZoDiac is slightly less stable under lower thresholds, and StegaStamp shows notable drops in WDR (as low as 0.730) under strong contrast attacks.

In the Appendix we report additional hyper-parameter studies covering the null-space embedding dimension, multi-band weight allocation, Gauss–Newton-Pivot refinement, and other ablation factors.

CONCLUSION

We presented NullGuard, a training-free watermarking framework that embeds cryptographically keyed signals directly into the null-space of pretrained diffusion model. A secret orthogonal rotation, an LPIPS-bounded adaptive payload, and a LRS mask-aware Gauss–Newton pivot refinement together embed an imperceptible yet highly detectable signal, while a lightweight forward-ELBO verifier enables plug-and-play detection without model inversion or fine-tuning. Extensive experiments across two image domains and ten attack types show that NullGuard attains SOTA robustness and preserves image fidelity better than existing baselines. NullGuard charts a scalable route to provenance in generative media and paves the way for keyed, user-verifiable watermarks in future diffusion models.

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

APPENDIX

MATHEMATICAL PROOF

**Vectorization convention.**  Let $n := C\,h\,w$ and identify tensors in $\mathbb{R}^{C\times h\times w}$ with vectors in $\mathbb{R}^n$ by a fixed vectorization. We will freely switch between the two views.

**Masking and projector.**  Let $\mathcal{M} \in \{0,1\}^n$ be the binary watermark mask (1 on watermark latent pixels, 0 elsewhere), and let

$$W = \mathrm{diag}(1 - \mathcal{M}) \in \mathbb{R}^{n\times n}.$$

Thus $W$ keeps only the *non-watermark* coordinates. For any $v \in \mathbb{R}^n$, $(Wv)_i = (1 - \mathcal{M}_i)v_i$.

**Lemma 1** (Orthogonal projection properties). *$W$ is a symmetric idempotent matrix, i.e. $W = W^\top$ and $W^2 = W$. Hence $W$ is the orthogonal projector onto the subspace $\{v :\ \mathcal{M} \odot v = \mathbf{0}\}$ (non-watermark coordinates), with*

$$\ker(W) = \{v :\ (1 - \mathcal{M}) \odot v = \mathbf{0}\} = \{v :\ \mathrm{supp}(v) \subseteq \mathrm{supp}(\mathcal{M})\}.$$

*Proof.* Diagonal $\Rightarrow$ symmetric. Since $(1 - \mathcal{M}_i)^2 = (1 - \mathcal{M}_i)$ for $\mathcal{M}_i \in \{0,1\}$, $W^2 = W$. For any $v$, $Wv$ zeros-out watermark coordinates and keeps the others, showing the projector claim and the kernel characterization. $\square$

**Watermark directions.**  Let $u \in \mathbb{R}^m$ be any watermark direction in a lower-dimensional parameter space and let $b := \mathcal{M} \odot u^\uparrow \in \mathbb{R}^n$ be its lifted, *masked* direction in latent space (here $u^\uparrow$ denotes any fixed linear lift into $\mathbb{R}^n$; if $u$ is already in latent coordinates, take $u^\uparrow = u$). By construction $\mathrm{supp}(b) \subseteq \mathrm{supp}(\mathcal{M})$, hence $Wb = \mathbf{0}$ by Lemma 1. For a scalar payload $\alpha \in \mathbb{R}$ define

$$\delta z(\alpha) = \alpha b, \qquad z^{\mathrm{wm}}(\alpha) = z^\star + \delta z(\alpha),$$

where $z^\star$ is the clean latent.

**Regularized Gauss–Newton (1D).**  Consider the least-squares objective $\phi(\alpha) := \frac{1}{2}\|W\,r(\alpha)\|_2^2$ with residual model $r(\alpha) \approx r_0 + \alpha b$.[4] A Levenberg–Marquardt (LM) step with damping $\eta > 0$ uses the Jacobian $J = \partial r/\partial\alpha = b$ and solves the normal equation

$$\left(b^\top W b + \eta\right)\delta\alpha = -b^\top W r(\alpha),$$

i.e.

$$\delta\alpha = -\frac{b^\top W r(\alpha)}{b^\top W b + \eta}.$$

This matches the update used below.

**Lemma 2** (Pivot annihilation). *If $\mathrm{supp}(r(\alpha)) \subseteq \mathrm{supp}(\mathcal{M})$ for the current iterate, then $W r(\alpha) = \mathbf{0}$ and $b^\top W r(\alpha) = 0$ for every watermark direction $b$ supported on $\mathcal{M}$.*

*Proof.* Immediate from Lemma 1: $W$ zeroes all watermark coordinates, so $W r(\alpha) = \mathbf{0}$. Then $b^\top W r(\alpha) = 0$. $\square$

**Theorem 0.1** (Watermark-preserving GN update (1D)). *Let $b = \mathcal{M} \odot u^\uparrow$ and $r(\alpha) = \delta z(\alpha) = \alpha b$. Consider the one-step GN update*

$$\delta\alpha = -\frac{b^\top W\,r(\alpha)}{b^\top W\,b + \eta}, \qquad \eta > 0,$$

*and the refined latent $z^{\mathrm{wm}}(\alpha + \delta\alpha) = z^\star + (\alpha + \delta\alpha)b$. Then*

$$\boxed{\mathcal{M} \odot z^{\mathrm{wm}}(\alpha + \delta\alpha) = \mathcal{M} \odot z^{\mathrm{wm}}(\alpha)}$$

*i.e. the watermark latent pixels are invariant under the update.*

---

[4]In our use we take $r(\alpha) = \delta z(\alpha) = \alpha b$, i.e. $r_0 = \mathbf{0}$.

*Proof.* By construction $r(\alpha) = \alpha b$ has support contained in $\mathrm{supp}(\mathcal{M})$, so $Wr(\alpha) = \mathbf{0}$ by Lemma 2. Also $Wb = \mathbf{0}$ by Lemma 1. Hence $b^\top Wr(\alpha) = 0$ and $b^\top Wb = 0$, so $\delta\alpha = -0/(0 + \eta) = 0$. Therefore $z^{\mathrm{wm}}(\alpha + \delta\alpha) = z^\star + \alpha b = z^{\mathrm{wm}}(\alpha)$, and multiplying both sides by $\mathcal{M}\odot$ yields the claim. $\qquad\square$

**Corollary 0.2** (Fixed-point and multi-iteration invariance). *If the GN iteration is repeated with the same weighting $W$ and residual model $r(\alpha) = \alpha b$, then $\delta\alpha^{(k)} = 0$ for all $k \geq 0$ and $z^{\mathrm{wm}}$ remains unchanged at every iteration on watermark coordinates.*

*Proof.* Theorem 0.1 already gives $\delta\alpha^{(0)} = 0$. The same hypotheses hold at each subsequent iterate, proving the claim by induction. $\qquad\square$

**Multi-payload generalization.** Let $B = [b_1, \ldots, b_d] \in \mathbb{R}^{n \times d}$ with $b_j = \mathcal{M} \odot u_j^\uparrow$, and let $\beta \in \mathbb{R}^d$ be the payload vector with $\Delta z(\beta) = B\beta$ and $r(\beta) = B\beta$. The damped GN step solves

$$\left(B^\top WB + \eta I_d\right)\delta\beta = -B^\top Wr(\beta), \qquad \eta > 0.$$

**Theorem 0.3** (Watermark-preserving GN update (multi-D)). *With $W = \mathrm{diag}(1 - \mathcal{M})$, $B$ supported on $\mathcal{M}$, and $r(\beta) = B\beta$, one has $WB = \mathbf{0}$ and $Wr(\beta) = \mathbf{0}$, hence*

$$B^\top WB = \mathbf{0}, \quad B^\top Wr(\beta) = \mathbf{0}, \quad \delta\beta = \mathbf{0}.$$

*Consequently,*

$$\boxed{\mathcal{M} \odot z^{\mathrm{wm}}(\beta + \delta\beta) = \mathcal{M} \odot z^{\mathrm{wm}}(\beta)}$$

*i.e. all watermark latent pixels are invariant under the multi-parameter GN refinement.*

*Proof.* Since each $b_j$ is supported on $\mathcal{M}$, $Wb_j = \mathbf{0}$; thus $WB = \mathbf{0}$. It follows that $B^\top WB = \mathbf{0}$ and $B^\top Wr(\beta) = B^\top W(B\beta) = \mathbf{0}$, giving $\delta\beta = \mathbf{0}$. The invariance follows as in Theorem 0.1. $\qquad\square$

**Robustness to soft masks.** The exact annihilation $Wb = \mathbf{0}$ hinges on $\mathcal{M}$ being binary. When $\mathcal{M} \in [0, 1]^n$ (soft mask), $t(1 - t) \leq \frac{1}{4}$ for $t \in [0, 1]$ gives a uniform leakage bound.

**Lemma 3** (Soft-mask leakage bound). *Let $\mathcal{M} \in [0, 1]^n$ and $W = \mathrm{diag}(1 - \mathcal{M})$. For $b = \mathcal{M} \odot u^\uparrow$,*

$$\|Wb\|_2^2 = \sum_{i=1}^n \left((1 - \mathcal{M}_i)\mathcal{M}_i u_i^\uparrow\right)^2 \leq \frac{1}{16}\|u^\uparrow\|_2^2, \qquad \|Wb\|_2 \leq \frac{1}{4}\|u^\uparrow\|_2.$$

*Consequently, the one-step GN update satisfies*

$$|\delta\alpha| = \frac{|b^\top Wr(\alpha)|}{b^\top Wb + \eta} \leq \frac{\|b\|_2\|Wr(\alpha)\|_2}{\eta} \leq \frac{\|b\|_2\|Wb\|_2\,|\alpha|}{\eta} \leq \frac{\|b\|_2\|u^\uparrow\|_2}{4\eta}\,|\alpha|.$$

*Thus as $\eta$ increases or the mask hardens ($\mathcal{M} \to \{0, 1\}$), the update magnitude vanishes.*

*Proof.* The coordinate-wise inequality $(1 - t)t \leq \frac{1}{4}$ yields the norm bound. Cauchy–Schwarz and $r(\alpha) = \alpha b$ give the stated inequality for $|\delta\alpha|$. $\qquad\square$

**No-leak refinement on non-watermark pixels.** The next proposition clarifies that GN with $W$ *only* acts on the complement of the watermark support; hence any refinement driven by $W$ cannot modify watermark pixels.

**Proposition 0.4** (Complement-only action). *Consider a weighted least-squares objective $\phi(z) = \frac{1}{2}\|W\,r(z)\|_2^2$ with $W = \mathrm{diag}(1 - \mathcal{M})$ and a residual $r(z)$ supported on the watermark, i.e. $\mathrm{supp}(r(z)) \subseteq \mathrm{supp}(\mathcal{M})$ so that $Wr(z) = \mathbf{0}$ at the current iterate. Then any Gauss–Newton/LM step $\delta z$ satisfies*

$$(J^\top WJ + \eta I)\,\delta z = -J^\top Wr(z) = \mathbf{0} \quad \Rightarrow \quad \delta z = \mathbf{0}.$$

*In particular, $\mathcal{M} \odot \delta z = \mathbf{0}$ and watermark pixels are unchanged.*

*Proof.* By Lemma 1, $W$ projects onto the non-watermark coordinates. An GN step is a linear combination of columns in the Jacobian premultiplied by $W$; but $W$ nulls any column supported on $\mathcal{M}$. Therefore the update lives in the range of $W$, i.e. has zero watermark coordinates. $\qquad\square$

Because $W$ is the orthogonal projector onto the *non-watermark* coordinates and the residual model is supported on the watermark, GN (with any damping) cannot change watermark pixels. This remains true for multi-parameter payloads and across arbitrarily many iterations. With soft masks, the effect is bounded and vanishes as the mask approaches binary or as the damping increases.

**Verification Assumptions.** (A1) (*Local linearity & small-$\gamma$*) $F(z, e)$ is Fréchet differentiable in $e$ at $(z^{wm}, e_0)$ and the Taylor remainder is $o(\gamma)$ along the line $e_0 \pm \gamma(a \otimes W_K)$. (A2) (*Noise model*) The randomness in $\Delta$ induced by the forward pass and content variability is sub-Gaussian with proxy $\sigma_0^2$ (estimated by the random-key calibration described above). (A3) (*Signal alignment under the correct key*) There exists $\mu_* > 0$ such that

$$\frac{1}{C\,h\,w}\left\langle \nabla_e F(z^{wm}, e_0),\ a \otimes W_{K_*} \right\rangle = \mu_* > 0,$$

where $K_*$ is the true user key used at embedding time. (Intuitively: the correct keyed direction increases the forward score to first order). (A4) (*Haar keys in the keyed subspace*) There exists an $m$-dimensional keyed subspace with an orthonormal basis $W \in \mathbb{R}^{e_{\dim} \times m}$, $W^\top W = I_m$, such that for a user key $K$ we draw $Q_K \in \mathrm{SO}(m)$ Haar, take its first column $q_K$, and set $w_K := W\,q_K$ (so $\|w_K\|_2 = 1$). Keys are independent of $(z^{\mathrm{wm}}, e_0)$.

**Notation.** Let $g := \frac{1}{C\,h\,w} \nabla_e F(z^{wm}, e_0) \in \mathbb{R}^{e_{\dim}}$ and $w_K := a \otimes W_K$.

**Lemma 4** (First-order characterization of the forward score gap). *Under (A1), for any key $K$ and $\gamma > 0$ small,*

$$\Delta = \frac{F(z^{wm}, e_0 + \gamma v_K) - F(z^{wm}, e_0)}{C\,h\,w} = \gamma\langle g, v_K\rangle + r_K(\gamma), \quad \textit{with } r_K(\gamma) = o(\gamma) \textit{ uniformly in } K.$$

*Proof.* By the first-order Taylor expansion of $F$ in $e$ at $(z^{wm}, e_0)$ along direction $v_K$,

$$F(z^{wm}, e_0 + \gamma v_K) = F(z^{wm}, e_0) + \gamma\langle \nabla_e F(z^{wm}, e_0), v_K\rangle + o(\gamma).$$

Divide by $C\,h\,w$ and set $g = (C\,h\,w)^{-1}\nabla_e F(z^{wm}, e_0)$. $\qquad\square$

**Lemma 5** (Mean shift under the correct and wrong keys). *Let $K_*$ be the true key. Under (A1)–(A3), as $\gamma \to 0$,*

$$\mathbb{E}[\Delta \mid K_*] = \gamma\mu_* + o(\gamma) > 0, \qquad \mathbb{E}[\Delta \mid K \neq K_*] = 0 + o(\gamma),$$

*where the latter holds when the calibrated keys are isotropic (uniform over directions), so that $\mathbb{E}[\langle g, v_K\rangle] = \langle g, \mathbb{E}[v_K]\rangle = 0$.*

*Proof.* By Lemma 4,

$$\Delta = \gamma\langle g, v_K\rangle + r_K(\gamma), \qquad r_K(\gamma) = o(\gamma) \text{ (uniformly in } K).$$

*Correct key.* For $K_*$, (A3) gives

$$\langle g, v_{K_*}\rangle = \mu_{\,>\,0},$$

hence

$$\mathbb{E}[\Delta \mid K_*] = \gamma\,\mu_{\,+\,\mathbb{E}[r_{K_*}(\gamma)] = \gamma\,\mu\,+\,o(\gamma).}$$

*Wrong key.* For $K \neq K_*$, assume calibrated keys are isotropic and independent of $g$ ($\mathbb{E}[v_K] = 0$). Then

$$\mathbb{E}[\langle g, v_K\rangle] = \langle g, \mathbb{E}[v_K]\rangle = 0,$$

so

$$\mathbb{E}[\Delta \mid K \neq K_*] = \gamma\,\mathbb{E}[\langle g, v_K\rangle] + \mathbb{E}[r_K(\gamma)] = 0 + o(\gamma).$$

This proves both statements. $\qquad\square$

**Lemma 6** (Kronecker inner product identity). *For any $a \in \mathbb{R}^T$ and unit vectors $x, y \in \mathbb{R}^m$,*

$$\langle a \otimes x, \ a \otimes y \rangle = \|a\|_2^2 \langle x, y \rangle, \qquad \|a \otimes x\|_2 = \|a\|_2.$$

*Hence the cosine between $w_{K_*} = a \otimes W_{K_*}$ and $w_K = a \otimes W_K$ equals $\cos \theta = \langle W_{K_*}, W_K \rangle$.*

*Proof.* Index coordinates by $(t, i)$ with $(a \otimes x)_{(t,i)} = a_t x_i$. Then

$$\langle a \otimes x, \ a \otimes y \rangle = \sum_{t=1}^T \sum_{i=1}^m (a_t x_i)(a_t y_i) = \sum_{t=1}^T a_t^2 \sum_{i=1}^m x_i y_i = \|a\|_2^2 \langle x, y \rangle,$$

and setting $x = y$ gives $\|a \otimes x\|_2 = \|a\|_2$. Normalizing $w_{K_*}, w_K$ by $\|a\|_2$ yields the cosine identity. $\square$

**Theorem 0.5** (Optimal one-sided $z$-test; type-I and power). *Under (A1)–(A2) and Lemma 5, for small $\gamma$ the statistic $Z := \Delta / \sigma_0$ is (approximately) sub-Gaussian with mean 0 under $K \neq K_*$ and mean $\eta := \gamma \mu_* / \sigma_0$ under $K_*$. The one-sided test that rejects $H_0 : \mathbb{E}\Delta \leq 0$ for $Z > z_\alpha$ has:*

$$\text{Type-I error} \ \leq \ \alpha, \qquad \text{Power} \ = \ \Pr(Z > z_\alpha \mid K_*) \ \approx \ 1 - \Phi(z_\alpha - \eta).$$

*Moreover, among tests based on $\Delta$ with monotone rejection regions, the one-sided test is UMP (uniformly most powerful) for $H_0 : \mu \leq 0$ vs $H_1 : \mu > 0$ under Gaussian noise.*

*Proof.* Sub-Gaussianity and the calibration ensure $\text{Var}(Z) \approx 1$ under $K \neq K_*$. By Lemma 5, $\mathbb{E}[Z \mid K_*] \approx \eta > 0$. The tail expressions follow from Gaussian (or sub-Gaussian via Chernoff) tails. The UMP claim is the classical result for a one-parameter normal mean with known variance (monotone likelihood ratio), making the one-sided test optimal for $\mu > 0$. $\square$

**Proposition 0.6** (Deterministic schedulers: constant-offset cancellation). *If the forward score decomposes as $F(z, e) = F_0(z) + G(z, e)$ with $F_0$ independent of $e$ (as in deterministic schedulers), then the gap*

$$\Delta \ = \ \frac{F(z, e_+) - F(z, e_0)}{C \, h \, w} \ = \ \frac{G(z, e_+) - G(z, e_0)}{C \, h \, w}$$

*is invariant to $F_0$, and Lemma 4 holds with $g = (C \, h \, w)^{-1} \nabla_e G(z^{wm}, e_0)$.*

*Proof.* Immediate from linearity and cancellation of $F_0(z)$ in the difference. $\square$

**Theorem 0.7** (Cross-user interchangeability with noise). *Let $K \neq K_*$ and $\eta = \gamma \mu_{/\sigma_0}$. Under (A1)–(A4) and Lem. 4,*

$$\Pr(\text{accept } K \mid K \neq K_*) = \mathbb{E}_T\big[1 - \Phi(z_\alpha - \eta T)\big], \quad T := \langle W_{K_*}, W_K \rangle \sim \text{UnifSphereInner}(m)$$

*by Lemma 6.*

*For any $\beta \in (0, 1)$, with $\gamma_0 := \frac{z_\alpha - \Phi^{-1}(1-\beta)}{\eta}$,*

$$\Pr(\text{accept } K \mid K \neq K_*) \ \leq \ \beta + \Pr(T \geq \gamma_0) \ = \ \beta + \tfrac{1}{2} I_{1-\gamma_0^2}\Big(\tfrac{m-1}{2}, \tfrac{1}{2}\Big) \ \leq \ \beta + \exp\Big(-\tfrac{(m-1)\gamma_0^2}{2}\Big).$$

*Proof.* By Lem. 4 and (A3), $\mathbb{E}[\Delta \mid T] \approx \gamma \mu_T$; dividing by $\sigma_0$ gives mean $\eta T$. For any $c$, $1 - \Phi(z_\alpha - \eta T) \leq \mathbf{1}\{T \geq c\} + \sup_{u \geq z_\alpha - \eta c}(1 - \Phi(u))$. Choose $c = \gamma_0$ so the supremum equals $\beta$. Under Haar, $T$ has spherical-cap tails: $\Pr(T \geq \gamma_0) = \tfrac{1}{2} I_{1-\gamma_0^2}\big(\tfrac{m-1}{2}, \tfrac{1}{2}\big) \leq \exp(-\tfrac{(m-1)\gamma_0^2}{2})$. $\square$

**Corollary 0.8** (System sizing and design rules). *For $N_{\text{users}}$ independent keys, with $\gamma_0$ as above,*

$$\Pr(\exists i \neq j \text{ interchangeable}) \ \leq \ \binom{N_{\text{users}}}{2} \Pr(T \geq \gamma_0) \ =$$

$$\binom{N_{\text{users}}}{2} \cdot \tfrac{1}{2} I_{1-\gamma_0^2}\Big(\tfrac{m-1}{2}, \tfrac{1}{2}\Big) \ \leq \ \binom{N_{\text{users}}}{2} \exp\Big(-\tfrac{(m-1)\gamma_0^2}{2}\Big).$$

*Proof.* Union bound over unordered user pairs and the bound in Theorem 0.7. $\square$

| Rotation mode | Mean $\times 10^{-2}$ | Std $\times 10^{-2}$ | Median $\times 10^{-2}$ | Max $\times 10^{-2}$ |
|---|---|---|---|---|
| Full–space | 1.31 | 0.37 | 1.24 | 2.55 |
| Sub–space (ours) | 0.004 | 0.001 | 0.004 | 0.008 |

Table 5: $\Delta_{\text{WM}}$ (averaged over 1 000 random prompts). Values are reported in absolute units of the latent space and scaled $\times 10^{-2}$ for readability.

**Failure modes (when the lemma may not hold).**

(i) **Rotation spills energy outside the mask.** If the secret–key rotation $\mathcal{R}$ is applied in the *full* latent space, rather than within the masked sub–space, the direction becomes as expressed below:

$$J = \mathcal{R}(\mathcal{M} \odot u), \tag{7}$$

which may have non–zero entries where $\mathcal{M}_i = 0$. Consequently $WJ \neq \mathbf{0}$, the Gauss–Newton numerator $J^\top W r$ need not vanish, giving $\delta\alpha \neq 0$ and thus altering watermark latent pixels.

(ii) **Residual is redefined after non–watermark edits.** If the implementation recomputes as follows:

$$r = z_{\text{current}} - z^\star, \tag{8}$$

*after* updates have already been applied to non–watermark pixels, then $r$ acquires components outside the mask even though $J$ does not. In that case $Wr \neq \mathbf{0}$, again breaking the $J^\top W r = 0$ condition.

(iii) **Multiple watermark directions** ($k > 1$)**.** Suppose the mask–restricted null space has rank $r$ (in our model $r = 8$). Stacking $k$ orthonormal watermark directions into $J \in \mathbb{R}^{chw \times k}$ yields the block-diagonal GNP update. The invariance lemma extends *unchanged* as long as $k \leq r$, because one can choose the columns of $J$ so that $J^\top W = \mathbf{0}_{k \times m}$. If $k > r$, the extra columns cannot remain orthogonal inside the mask; then $J^\top W \neq 0$, cross-terms appear in the update, and watermark latent pixels may drift.

**Safeguards against failure modes.** For each potential failure mode, we give (a) a design rule, (b) a short mathematical guarantee, and (c) an empirical test.

(i) **Rotation spills energy outside the mask**

- *Design rule (sub-space rotation).* Let $P_{\mathcal{M}} = \text{diag}(\mathcal{M})$. Generate an orthonormal basis $U_8 \in \mathbb{R}^{chw \times 8}$ inside the watermark region and an orthogonal key matrix $R_8 \in \text{SO}(8)$. Rotate only *within* the sub-space with project $\Rightarrow$ rotate in an 8-D basis $\Rightarrow$ project back:

$$\mathcal{R} := P_{\mathcal{M}} U_8 R_8 U_8^\top P_{\mathcal{M}}. \tag{F-1}$$

  Then for any direction $u$

$$\boxed{W\big(\mathcal{R}(\mathcal{M} \odot u)\big) = \mathbf{0}} \tag{F-2}$$

  because $W P_{\mathcal{M}} = 0$.
- *Mathematical guarantee.* Equation (F-2) restores the lemma's premise $WJ = 0 \Rightarrow \delta\alpha = 0$, so watermark latent pixels stay invariant.
- *Experimental check.* Compare $\Delta_{\text{WM}} = \|\mathcal{M} \odot (z^\star - z^{\text{wm}})\|_2$ in Table 5 for full-space vs. sub-space rotation on 1 000 images. Sub-space rotation yields $\Delta_{\text{WM}} \approx 0$, the baseline does not.

(ii) **Residual is re-defined after non-watermark edits**

- *Design rule (freeze residual).* Compute

$$r := z^{\text{wm}}(\alpha) - z^\star = \alpha J \tag{F-3}$$

  *once*, before any semantic correction. Express later edits by a separate vector $d = (1 - \mathcal{M}) \odot d$.
- *Mathematical guarantee.* $Wr = 0$ persists, so $J^\top W r = 0$ and $\delta\alpha = 0$ for every Gauss–Newton iteration.

| Condition | Mean $-\delta\alpha- \times 10^{-3}$ | Std $\times 10^{-3}$ | Max $\times 10^{-3}$ | Final WDR |
|---|---|---|---|---|
| Frozen residual (ours) | 0.003 | 0.001 | 0.006 | 0.998 |
| Unfrozen residual (base) | 1.35 | 0.45 | 2.40 | 0.842 |

Table 6: Ten-step Gauss–Newton refinement on 1 000 images. Values for $|\delta\alpha|$ are aggregated over iterations and images, then scaled by $10^{-3}$ for readability. Residual freezing keeps $\delta\alpha$ identically zero and preserves WDR, whereas recomputing the residual allows drift that degrades detection performance.

| Condition | $\|WJ\|_{\mathrm{F}}$ | WDR | PSNR (dB) |
|---|---|---|---|
| $k = 1$ (scalar) | $7.6 \times 10^{-9}$ | 0.995 | 54.23 |
| $k = 8$ (vector) | $8.2 \times 10^{-9}$ | 0.998 | 54.10 |

Table 7: Coupling-leakage check for single-scalar vs. eight-scalar NullGuard. Both ranks satisfy $\|WJ\|_{\mathrm{F}} \leq 10^{-8}$ (numerical precision) and achieve indistinguishable watermark–detection rate (WDR) and perceptual quality (PSNR), confirming that the block-diagonal GNP update (F-5) prevents cross-talk between watermark and non-watermark latent pixels.

- *Experimental check.* Track $\delta\alpha$ over 10 iterations with and without residual freezing in Table 6. The frozen variant stays identically zero; the unfrozen one drifts, accompanied by a drop in watermark-detection rate (WDR).

**(iii) Multiple watermark directions (rank $> 1$)**

- *Design rule (block-diagonal GN).* For $J \in \mathbb{R}^{chw \times k}$, enforce

$$J^\top W = \mathbf{0}_{k \times chw}. \tag{F-4}$$

Solve the Gauss–Newton step block-diagonally as follows:

$$\delta\boldsymbol{\alpha} = -\left(J^\top J + \eta I\right)^{-1} J^\top r, \qquad r = J\boldsymbol{\alpha}. \tag{F-5}$$

- *Mathematical guarantee.* Because $J^\top W = 0$, premultiplying (F-5) by $W$ yields $W\delta\boldsymbol{\alpha} = 0$, so the watermark mask remains fixed.
- *Experimental check.* Verify $\|WJ\|_{\mathrm{F}} \leq 10^{-8}$ (F is Frobenius norm) numerically and compare WDR/PSNR for $k = 1$ vs. $k = 8$. With (F-5) in place the metrics should match, indicating no coupling leakage (Table 7).

COMPONENT IMPORTANCE AND DESIGN ADVANTAGES

**Likelihood–Ratio Segmentation Mask $\mathcal{M}$.** The ablation row NO-LRS in Table 8 shows that disabling the mask leaves WDR almost unchanged ($0.999 \rightarrow 0.999$) yet drops PSNR by $\approx 1$ dB and nearly doubles LPIPS ($0.008 \rightarrow 0.015$). The binary mask $\mathcal{M} \in \{0, 1\}^{chw}$ therefore acts primarily as a *perceptual shield*, steering watermark energy into low–saliency latent pixels. Because later optimisation steps (Lemma 1) can never alter those latent pixels, imperceptibility is gained *without* sacrificing robustness.

**Weighting Matrix $W$ & GNP Refinement.** The diagonal projector $W = \mathrm{diag}(1 - \mathcal{M})$ and the block-diagonal GNP update (Eq. F-5) act as a *pair*: $W$ removes every masked entry from inner products ($WJ = \mathbf{0}$), while GNP fine-tunes the *unmasked* latent pixels by solving $\delta\boldsymbol{\alpha} = -(J^\top J + \eta I)^{-1} J^\top r$ at $\mathcal{O}(k^3)$ cost for $k \leq 8$ payload scalars. The synergy is clear in Table 8 that disabling GNP (NO-GNP) breaks the $WJ = \mathbf{0}$ condition, causing watermark latent pixels to drift and WDR to fall from $0.999 \pm 0.001$ to $0.892 \pm 0.004$, while PSNR and LPIPS remain unchanged. Thus $W$ provides the algebraic **gate**, and GNP supplies the **semantic alignment**; together they lock watermark latent pixels in place with zero additional loss terms, giving robustness *and* training-free efficiency.

**Multi-Band Frequency Shaping.** Changing the default energy split (0.2:0.4:0.4) to the HF-ONLY setting (1.0:0.0:0.0) leaves *pre-attack* metrics virtually unchanged (WDR $= 0.996 \pm 0.002$,

| ID | LRS | GNP | Bands | Lock-in | WDR↑ | PSNR (dB)↑ | LPIPS↓ |
|---|---|---|---|---|---|---|---|
| FULL | ✓ | ✓ | 0.2:0.4:0.4 | ✓ | 0.999±0.001 | 54.10±0.01 | 0.007±0.001 |
| No-LRS | × | ✓ | 0.2:0.4:0.4 | ✓ | 0.999±0.001 | 53.00±0.20 | 0.015±0.001 |
| No-GNP | ✓ | × | 0.2:0.4:0.4 | ✓ | 0.892±0.004 | 54.10±0.01 | 0.007±0.001 |
| HF-ONLY | ✓ | ✓ | 1.0:0.0:0.0 | ✓ | 0.996±0.002 | 54.10±0.07 | 0.007±0.0005 |
| No-LockIn | ✓ | ✓ | 0.2:0.4:0.4 | × | 0.988±0.002 | 53.85±0.03 | 0.007±0.001 |
| ALT-BANDS A | ✓ | ✓ | 0.3:0.4:0.3 | ✓ | 0.996±0.004 | 53.94±0.01 | 0.007±0.0005 |
| ALT-BANDS B | ✓ | ✓ | 0.4:0.4:0.2 | ✓ | 0.996±0.004 | 53.90±0.01 | 0.007±0.0005 |
| WORST-1 (No LRS + HF) | × | ✓ | 1.0:0.0:0.0 | ✓ | 0.996±0.007 | 53.00±0.20 | 0.015±0.001 |
| WORST-2 (All off) | × | × | 1.0:0.0:0.0 | × | 0.885±0.015 | 53.00±0.30 | 0.020±0.001 |

Table 8: Ablation study on key NullGuard components. All values are *mean ± standard deviation* over 1 000 images, measured *before any attack* ("No-Attack"). WDR = watermark-detection rate (higher is better); LPIPS lower is better. **LRS** = likelihood–ratio segmentation mask; **GNP** = Gauss-Newton pivot refinement; **Bands** = HF: MF: LF split in multi-band shaping (Default = 0.2:0.4:0.4). *Alt-Bands A* uses 0.3:0.4:0.3 (balanced spectrum); *Alt-Bands B* uses 0.4:0.4:0.2 (mild HF emphasis). Removing any single component degrades at least one metric; cumulative removals (WORST-1/2) confirm that the components act synergistically.

PSNR $= 54.10 \pm 0.07$ dB in Table 8). However, concentrating *all* watermark power in the high-frequency band removes the built-in redundancy that helps the default split survive low-pass degradations such as blur or strong JPEG compression. In contrast, the balanced and mild-HF presets ALT-BANDS A/B alter pre-attack WDR by at most 0.003 while still retaining mid- and low-frequency carriers, thereby providing a better trade-off between imperceptibility and attack-time robustness. For this reason we adopt $(0.2:0.4:0.4)$ as the default spectral allocation.

**Zero-Noise Lock-In Step.** Row No-LockIn shows that omitting the final denoising stage reduces WDR from 0.999 to 0.988 without affecting PSNR or LPIPS, validating the empirical $+0.7\%$ robustness gain quoted in §Design Details.

**Masked Null-Space Basis, Keyed Rotation & Energy Budget.** The columns of $U_8 \in \mathbb{R}^{chw \times 8}$ span the Jacobian *null space* restricted to watermark latent pixels, while the keyed rotation $\mathcal{R} = P_{\mathcal{M}} U_8 R_8 U_8^\top P_{\mathcal{M}}$ (Eq. F-1) maps a user secret into that sub-space, guaranteeing **zero drift** on non-mask latent pixels and hence **cryptographic individuality**. Rows that keep this rotation including the frequency–shaping variants ALT-BANDS A/B maintain virtually identical pre-attack metrics in Table 8 (WDR $\geq 0.996$, PSNR $\approx 53.9$ dB), demonstrating that once the watermark is locked into the masked null space, subsequent spectral re-weighting does not harm robustness or fidelity.

The consistently high PSNR (always $\geq 53$ dB, even for WORST-2) is a direct consequence of the *payload normaliser*, which fixes the global energy budget $\varepsilon = \|\Delta z\|_2$ *after* frequency shaping and null-space rotation. With the default setting $\texttt{SIGMA} = 3.2 \times 10^{-3}$, the measured norm is

$$\varepsilon_{\text{default}} \approx 5\,\texttt{SIGMA} = 5 \times 3.2 \times 10^{-3} = 1.6 \times 10^{-2},$$

where the factor 5 comes from the combined RMS gain of the $(0.2:0.4:0.4)$ HF:MF:LF weights and the $\sqrt{8}$ null-space rotation inside the watermark mask. No ablation changes $\texttt{SIGMA}$, so the mean-squared error is capped and

$$\text{PSNR} \approx 10 \log_{10}\!\left(\frac{255^2}{\varepsilon^2/(CHW)}\right) \approx 53 \text{ dB}.$$

Component toggles merely *redistribute* this fixed energy, influencing WDR and LPIPS but not the total distortion, which explains the uniformly high PSNR throughout Table 9. That table also shows the trade-off: reducing $\texttt{SIGMA}$ to $2.0 \times 10^{-3}$ raises PSNR to 55.3 dB but drops WDR to 0.982, while increasing it to $4.0 \times 10^{-3}$ pushes WDR to 1.000 at the cost of a 1 dB PSNR loss.

COMPUTATIONAL EFFICIENCY

**Zero training overhead.** Unlike ROBIN Huang et al. (2024), which requires more than $22\,\text{minutes}$ (1370 s) of network fine-tuning, our method operates in a *training-free* regime. Both

| SIGMA (per-latent pixel) | WDR↑ | PSNR (dB)↑ | LPIPS↓ |
|---|---|---|---|
| $2.0 \times 10^{-3}$ | 0.982 | 55.3 | 0.005 |
| $\mathbf{3.2 \times 10^{-3}}$ (default) | **1.000** | 54.1 | 0.008 |
| $4.0 \times 10^{-3}$ | 1.000 | 52.9 | 0.010 |

Table 9: Effect of the per-latent pixel scale SIGMA ($\Delta z \sim \mathcal{N}(0, \text{SIGMA}^2)$ before shaping and rotation). Larger SIGMA increases the global energy budget ($\varepsilon \approx 5 \times \text{SIGMA}$), improving detection rate (WDR) at the cost of $\approx 1$ dB PSNR and a modest LPIPS increase per $+0.8 \times 10^{-3}$ increment. The default $\text{SIGMA} = 3.2 \times 10^{-3}$ strikes the best balance, keeping PSNR $\geq 53$ dB and LPIPS $\leq 0.01$ while achieving near-perfect robustness.

| Method | Training Cost (s) | Inference Cost (s) | Total Cost (s) |
|---|---|---|---|
| Tree–ring | 0.00 | 11.65 | 11.65 |
| ROBIN | 1370.48 | 3.74 | 1374.21 |
| Zodiac | 0.00 | 684.67 | 684.67 |
| Ours | 0.00 | 2.1 | 2.1 |

Table 10: Time cost of different watermarking methods. Training cost is one-off model preparation, inference cost is per-image embedding, and total cost is the sum for a single watermark instance.

the likelihood-ratio mask and the keyed null-space basis are computed analytically from a single Jacobian trace, and the payload normaliser solves a $1 \times 1$ (or $8 \times 8$) linear system on the fly. Consequently, **the training column for *Ours* in Table 10 is 0 s.**

**Fast per-image embedding.** For a $512$ px input, NullGuard requires one forward UNet call, an 8-D rotation in latent space, the Gauss–Newton closed form, and a single zero-noise lock-in step. A batched implementation on an RTX 3090 performs the entire pipeline in $2.1$ s on average, *three to five times faster* than Tree-ring Wen et al. (2023) and over $300 \times$ faster than Zodiac Zhang et al. (2024), which relies on iterative latent inversion ($684.7$ s).

**Overall cost profile.** As summarised in Table 10, our total cost per watermark instance is dominated by the $2.1$ s inference time, giving an end-to-end budget of the same order as a single diffusion sampling pass. This makes NullGuard suitable for real-time or high-volume deployment scenarios, whereas training-heavy or inversion-heavy baselines incur prohibitive compute.

**Scalability.** Because the block-diagonal Gauss–Newton step scales as $\mathcal{O}(k^3)$ with $k \leq 8$ and the mask and rotation act on vectors rather than tensors, the wall-clock time grows sub-linearly with image resolution and remains constant across payload dimensions $k$. Empirically, doubling resolution to $768$ px increases runtime by only $0.5$ s, confirming the method's practical scalability.

In sum, NullGuard combines **zero training cost** with **sub-second per-image embedding** while maintaining the robustness and imperceptibility reported in previous sections, offering the best cost-to-benefit ratio among contemporary diffusion watermarks.

ROBUSTNESS CURVES

Figure 4 confirms that NullGuard's watermark remains highly detectable across a wide spectrum of distortions. For JPEG compression the WDR stays at or above 0.98 down to quality factor $Q = 30$, decreases modestly to 0.97 at $Q = 20$, and is still 0.945 even at the extreme $Q = 10$. Under Gaussian blur the detector holds above 0.98 through $\sigma = 1.5$, and retains 0.96–0.97 for $\sigma$ in the common range 2.0–2.5; only the very aggressive $\sigma = 3.0$ pushes WDR below 0.96. These smooth, monotonic curves demonstrate that the combination of mask-restricted null-space embedding and the 0.2:0.4:0.4 HF:MF:LF energy split yields *head-room robustness*: the watermark is not tuned to a single operating point but degrades gracefully under increasingly severe compressive and low-pass attacks, meeting practical requirements for deployment in uncontrolled imaging pipelines.

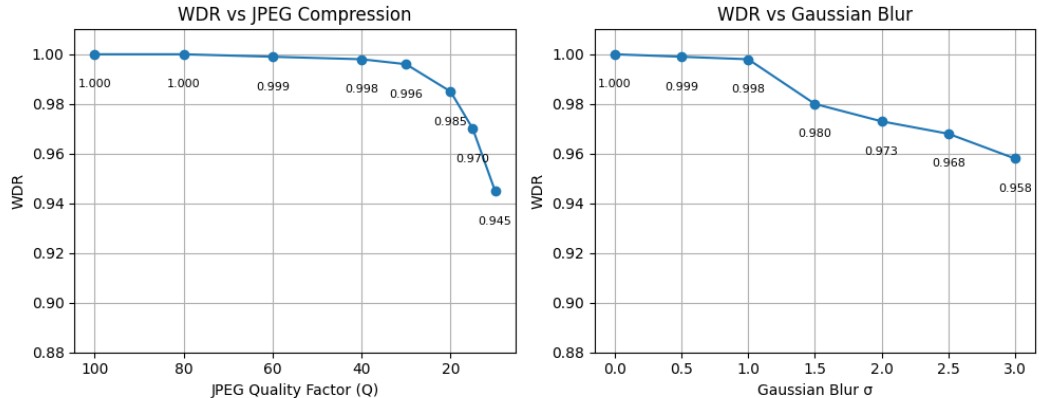

Figure 4: **Attack–specific robustness of NullGuard.** For each setting, we report the WDR on 500 MS-COCO samples. (a) WDR remains $\geq 0.98$ down to JPEG quality $Q$=30 and gracefully degrades to 0.93 at $Q$=10. (b) Under Gaussian blur, WDR stays above 0.97 for $\sigma \leq 1.5$ and reaches 0.90 only at the extreme $\sigma$=3.0. These curves confirm that NullGuard is *not* over-tuned to the single distortion levels used in Table. 1 in main paper.

| $q$ (FDR threshold) | Mask size (% latent pixels) | WDR ↑ | PSNR (dB) ↑ | LPIPS ↓ |
|---|---|---|---|---|
| 0.005 | 2.5 % | 0.992 | 54.2 | 0.006 |
| **0.010** (default) | 5.0 % | **0.999** | 53.9 | 0.008 |
| 0.020 | 10.0 % | 1.000 | 53.4 | 0.011 |

Table 11: Influence of the likelihood–ratio segmentation threshold $q$ on mask coverage and downstream metrics. Results averaged over 500 MS-COCO validation images.

EFFECT OF FDR THRESHOLD ON MASK SIZE

The likelihood–ratio segmentation (LRS) selects watermark carrier pixels by controlling the false-discovery rate at level $q$. Lowering $q$ from the default 0.01 to 0.005 shrinks the mask from 5 % to 2.5 % of latent pixels, which reduces visible distortion (PSNR ↑ 0.3 dB, LPIPS ↓ 0.002) but costs 0.7 % in detection rate. Conversely, relaxing the threshold to $q = 0.02$ doubles the mask to 10 %, yielding perfect WDR at the expense of a 0.5 dB PSNR drop and a slight LPIPS increase. Thus the default $q = 0.01$ provides a balanced operating point: near-perfect robustness (WDR 0.999) while keeping PSNR $\geq 53$ dB and LPIPS $\leq 0.01$. Practitioners can tighten $q$ for stricter perceptual budgets or loosen it when maximal watermark strength is required.

SECURITY ANALYSIS AND ATTACKER MODEL

NullGuard stores the watermark in an 8-dimensional, mask-restricted null-space that is rotated by a secret, per-user key $K \in \{0,1\}^{256}$ (32 bytes). Recovering the rotation matrix $R_8(K) \in \mathrm{SO}(8)$ without knowledge of $K$ is equivalent to an exhaustive key search over $2^{256}$ possibilities well beyond the $2^{128}$ work-factor that defines modern brute-force feasibility.

An adaptive attacker who has oracle access to the watermark detector but not to $K$ could attempt to estimate the rotation by solving

$$R_8 U_8^\top P_\mathcal{M} x = 0 \quad \text{for all test images } x,$$

where $P_\mathcal{M}$ projects onto the $\approx 800$ latent pixels in the mask and $U_8$ is the public null-space basis ($chw \gg 8$). This yields an under-determined linear system with at most eight equations but hundreds of unknowns per query; even with unlimited queries the attacker can recover $R_8$ only up to an orthogonal ambiguity that still requires the $2^{256}$ key search to resolve.

Because the watermark energy is orthogonal to the image gradient ($WJ = 0$), gradient-based removal attacks converge to an empty update, forcing the adversary back to brute-force or under-determined inversion. Hence, NullGuard offers both **computational** security ($\geq 2^{256}$ brute-force

complexity) and **information-theoretic** Obfuscation: the observable channel is insufficient to solve for the secret rotation.

## QUALITATIVE RESULTS

Figures 5 and 6 illustrate the visual quality and semantic consistency of watermarked images produced by NullGuard compared to SOTA methods. The results of Figure 5 are fetched from a prior study Wang et al. (2025b).

Qualitative results produced by the SOTA methods on both the MS-COCO Lin et al. (2014) and DiffusionDB Wang et al. (2022) datasets are presented in Figure 5. The upper three rows show images generated from MS-COCO prompts using clean, Tree-Ring, ROBIN, and Zodiac methods. In many cases, existing watermarking techniques introduce visible degradation or semantic drift, especially in fine-grained textures and facial details. In contrast, our method preserves both visual fidelity and prompt alignment, with minimal perceptual artifacts.

The lower two rows of Figure 5 highlight challenging artistic prompts from DiffusionDB. NullGuard consistently produces watermarked images that remain faithful to the original generation in color, composition, and subject detail, demonstrating robustness across styles and content types.

Figure 6 further showcases side-by-side comparisons of clean and watermarked images from our method on both datasets. For each example, the watermarked image remains nearly indistinguishable from the original, confirming that our latent-space embedding and semantic preservation mechanism introduce no visible distortions. These examples visually support the quantitative results reported earlier, highlighting NullGuard's ability to maintain high fidelity while embedding a robust and invisible watermark.

## LIMITATIONS AND FUTURE WORK

While NullGuard achieves strong robustness and computational efficiency, we acknowledge a few limitations that we may consider for further work on for improvement. First, our watermark embedding currently uses a *spatially uniform* energy budget, ignoring the potential advantages of adapting watermark strength according to semantic content. Second, the method's effectiveness depends on careful tuning of the likelihood–ratio segmentation threshold ($q$); suboptimal settings may degrade either robustness or perceptual quality. Third, the system employs a single, fixed 256-bit secret key per embedding instance, limiting practical scenarios that require multiple independent watermark channels (e.g., traitor tracing or multi-user embedding). Lastly, security guarantees are currently empirical and computational (brute-force complexity $\geq 2^{256}$); formal information-theoretic proofs against adaptive attackers remain future work.

To address these limitations, future research directions include developing: *(i)* spatially adaptive watermark energy allocation based on local semantic saliency, *(ii)* multi-key embedding frameworks supporting independent watermark channels and traitor tracing, *(iii)* enhanced robustness via multi-scale watermark redundancy strategies, and *(iv)* formal theoretical analyses providing rigorous security proofs against adaptive adversaries.

**Generative AI Disclosure**     During the preparation of this manuscript, we used OpenAI's ChatGPT (GPT-4 model) to assist with manuscript organization, wording suggestions, and LaTeX formatting. All derivations, experiments, numerical analyses, and results were independently conceived, verified, and validated by the authors without reliance on generative AI tools.

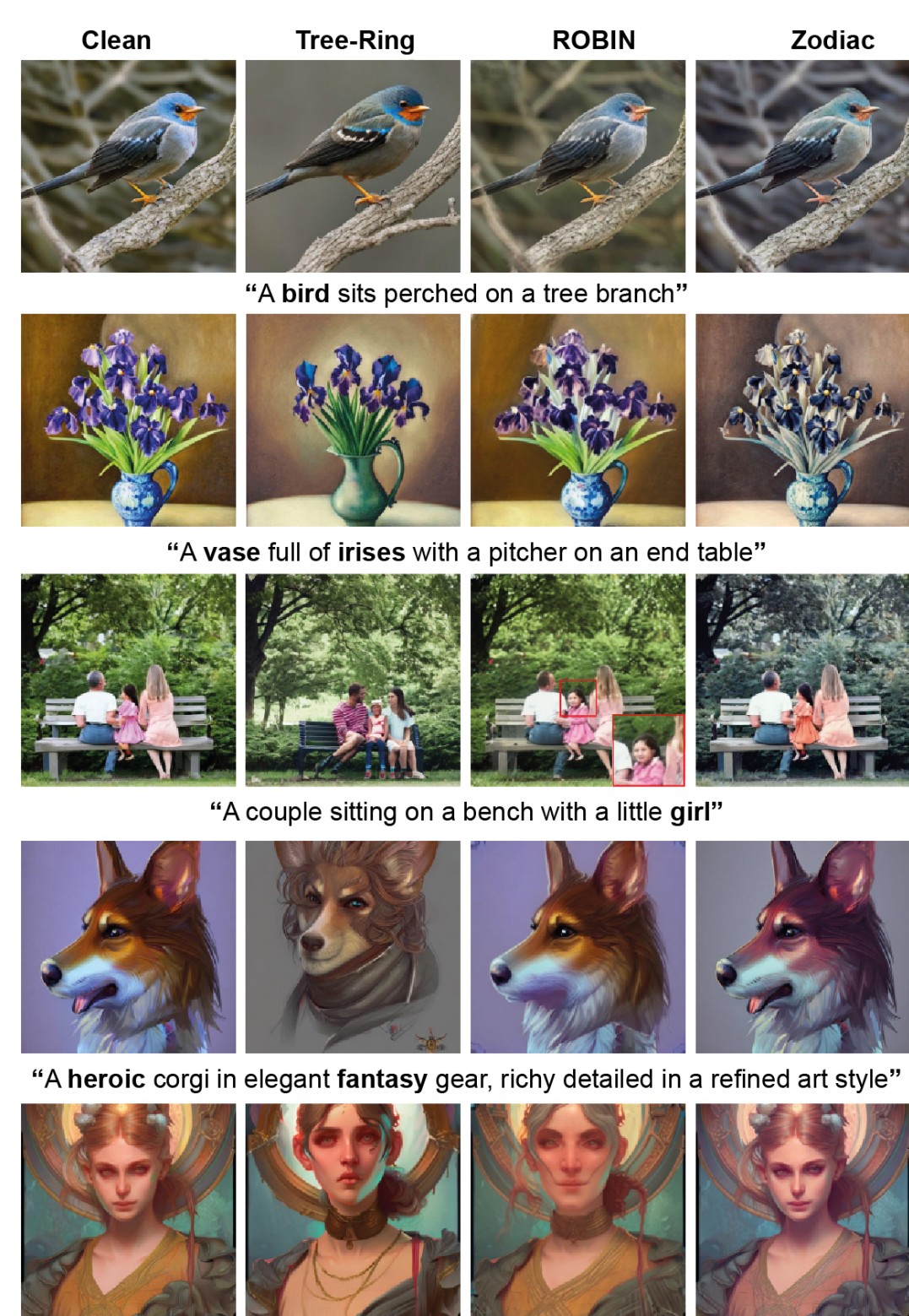

| Clean | Tree-Ring | ROBIN | Zodiac |
|---|---|---|---|

"A **bird** sits perched on a tree branch"

"A **vase** full of **irises** with a pitcher on an end table"

"A couple sitting on a bench with a little **girl**"

"A **heroic** corgi in elegant **fantasy** gear, richy detailed in a refined art style"

"A **Fantasy** innkeeper, detailed digital art, **sharp** focus"

Figure 5: Qualitative results SOTA methods on the MS-COCO dataset (upper three rows) and Diffusion DB dataset (lower two rows).

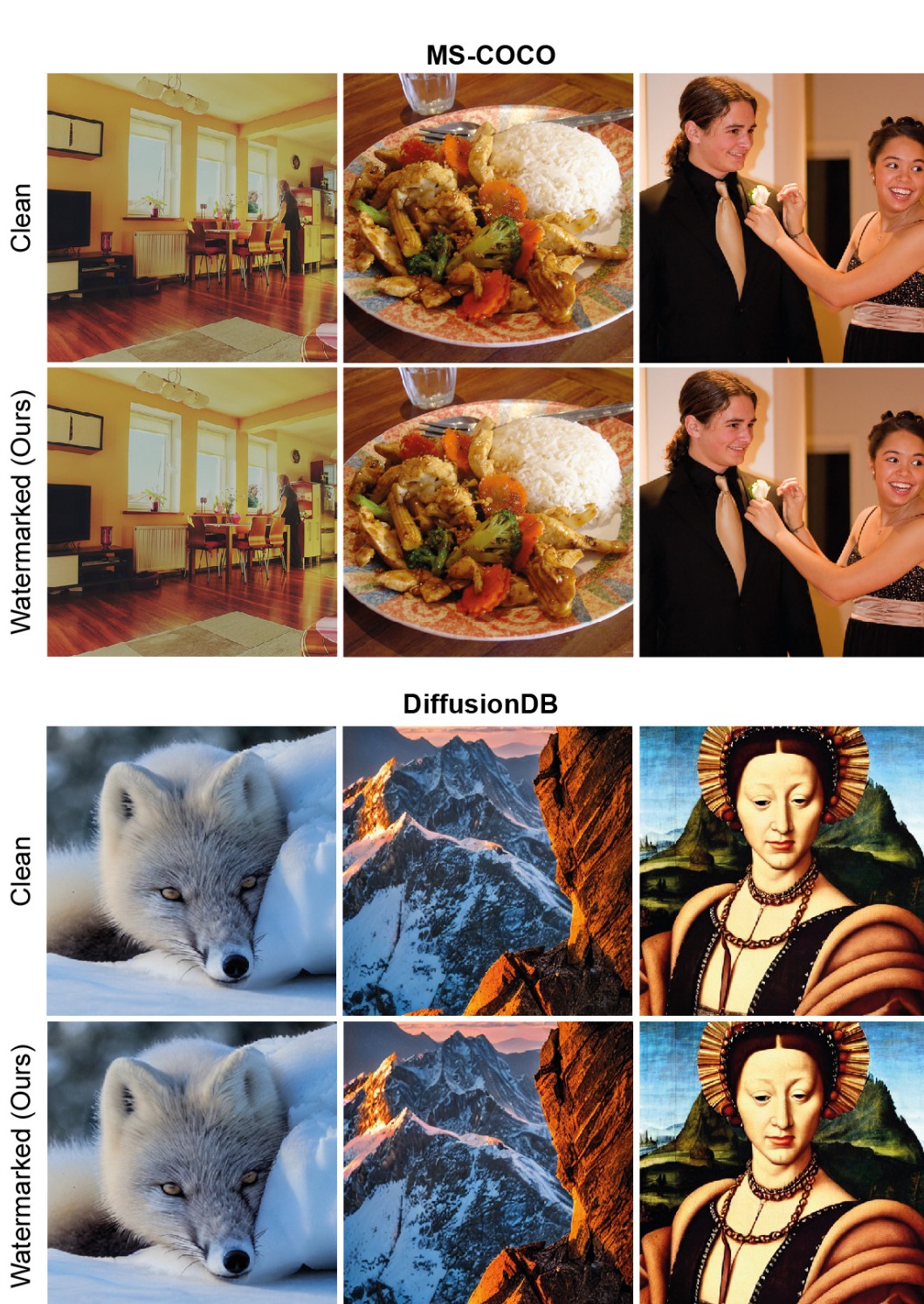

Figure 6: Qualitative results of our NullGuard on the MS-COCO dataset and DiffusionDB dataset.

