# OpenReview forum: "NullGuard: Null-Space Embedding for Driftless Invisible Image Watermarking"
_ICLR.cc/2026/Conference — ICLR 2026 Conference Withdrawn Submission_

### Official Review · Reviewer_LeHZ · 2025-10-14

**Soundness:** 2
**Presentation:** 1
**Contribution:** 2
**Rating:** 2
**Confidence:** 4

**Summary:**

This paper proposes NullGuard, a novel watermarking framework designed to embed robust and undetectable signals into images. The approach focuses on leveraging the null space of deep representations to insert watermark keys that do not interfere with the visible output but can be later retrieved. The framework is claimed to improve upon existing watermarking methods like Zodiac, offering better imperceptibility and robustness across a range of generative model tasks and perturbation settings.

**Strengths:**

The paper explores a promising direction in watermarking by leveraging the null space of internal representations in generative models, which could in theory allow embedding information without altering the observable output. The proposed method appears to achieve strong empirical results, and the evaluation follows standard benchmarks, including robustness to various common attacks.

**Weaknesses:**

The most significant issue with this paper lies in its writing quality and presentation, which severely hinders comprehension of both the technical method and its motivation. The clarity, structure, and detail of the method section are insufficient to support a confident understanding or reproduction of the proposed framework. Many core terms are undefined, motivations are vague or missing, and technical procedures are explained in a way that is too abstract or obscure.

First, there are persistent formatting and grammatical issues throughout the paper. The citation style is inconsistent with standard formatting, section and subsection headings are not numbered or inconsistently styled (e.g., Line 240), and numerous sentences are difficult to parse due to typos or awkward phrasing—for instance, Line 345’s "within," and Line 481’s "Once in To". Acronyms are introduced without definition, and paragraph transitions are often abrupt. These issues alone detract from the professional presentation expected at this level.

More importantly, the methodology section lacks a clear and logical narrative structure. A strong methods section should begin by providing a high-level overview of the pipeline: how the watermark is generated, how it is embedded, and how it is detected. Each of these stages should then be broken down into smaller components, with clear explanations of what each step is doing, why it is necessary, and how it contributes to the overall goal.

Taking “Key-to-Secret Rotation” as an example (Line 192), the reader is suddenly introduced to a “high-entropy matrix” without any prior definition or intuition. Why is the key K expanded into such a matrix? What is the purpose of performing a QR decomposition, and what is the significance of the resulting matrix belonging to the SO(m) group? What does “membership in SO(m)” even mean in this context? These are not rhetorical questions—they are necessary clarifications the paper does not provide. This lack of grounding occurs repeatedly throughout the method section. Each component (including the Null Space Embedder and the Decision-based Verifier) is introduced as if the reader is already familiar with the mathematical background and the reasoning behind each design choice, yet the necessary preliminaries and intuitions are entirely missing. The result is that the framework feels like a sequence of obscure technical operations without a clearly communicated purpose or justification.

Furthermore, the experimental section fails to analyze the method in a meaningful way. While the setup and baseline results are similar to those in the ZoDiac paper—an acceptable starting point—the paper does not offer any in-depth ablation, parameter sensitivity, or justification for why this proposed method works better. For a paper introducing multiple new modules, one would expect to see ablation experiments showing the relative contribution of each, or at the very least, tuning analyses of any unique hyperparameters. Yet the experiments simply report improvements and repeat the phrase “consistently outperform” without delving into why that might be the case. As a result, the experimental section serves more as a scoreboard than a validation of the proposed design.

To be clear, if the method section had been written clearly and the components had been strongly motivated, the lack of ablation might be excusable. But in the current form, where the methodology is opaque and underspecified, the lack of experimental introspection compounds the difficulty in evaluating the actual merit of the method.

**Questions:**

Please refer to the weakness part.

---

### Official Review · Reviewer_kwbP · 2025-10-26

**Soundness:** 3
**Presentation:** 3
**Contribution:** 2
**Rating:** 2
**Confidence:** 5

**Summary:**

This paper proposes an invisible watermarking technique named NullGuard, designed using the T2I-Diffusion model. Its purpose is to address the semantic shift issues caused by watermark embedding in Zodiac and a series of generative watermarking algorithms. NullGuard's core innovation lies in embedding the watermark signal within a precomputed, universal semantic null space. This framework is training-free and plug-and-play, featuring efficient validation through a novel “forward score gap” statistical test that eliminates the need for time-consuming DDIM inversion.

**Strengths:**

1.	Defining a “semantically orthogonal” watermark embedding space using the null space of the Jacobian matrix in diffusion models provides a robust theoretical foundation and stands in stark contrast to existing approaches (such as those based on fine-tuning or simple frequency domain/subspace operations).
2.	This solution requires no additional training and is plug-and-play, offering significant advantages over competing approaches by saving computational resources and time in fine-tuning diffusion models.
3.	This paper is well-written and easy to understand.

**Weaknesses:**

1.	The experimental evidence is insufficient. Typically, image watermarks require robustness not only against the distortion caused by the series of numerical modifications demonstrated in the paper, but also against geometric distortion caused by operations such as cropping, scaling, and perspective transformations. The authors only conducted experiments with a fixed rotation angle of 90 degrees.

2.	The paper's motivation and experiments are mismatched. In the Introduction, the authors state that the paper “addresses the challenge of semantic drift in diffusion-based watermarking by introducing a latent-space embedding strategy that minimizes modification to the semantic content of the image.” However, in the experimental section, the authors primarily compare their approach against classical post-processing watermarking schemes. Only ZoDiac is associated with generative models, while classic generative watermarking schemes like Tree-ring and Gaussian-shading are not evaluated in the experiments.
3.	The baseline did not incorporate recent work; aside from ZoDiac, the most recent studies were CIN and SSL from 2022, which lack persuasive power.

**Questions:**

In addition to the shortcomings raised in the Weakness section, I have several questions I would like the author to address.

1.	The author only presented images containing watermarks without revealing the distribution pattern of the watermarks. The author may later reference patterns such as Residual=scale_factor*(Encoded-Original).clamp() to demonstrate the watermarking scheme.

2.	I have a question regarding the verification method used by the author: if the image submitted for verification lacks a watermark, could this lead to misclassification due to the differing e0 and ek values used in the DDIM inverse? Such an obvious attack method compromises the reliability of the watermarking system.

---

### Official Review · Reviewer_sCbe · 2025-11-01

**Soundness:** 3
**Presentation:** 3
**Contribution:** 3
**Rating:** 6
**Confidence:** 3

**Summary:**

This paper proposes Nullgard, a training free invisible watermarking framework that leverages diffusion models. At its basis, it embeds a signal into the null space of diffusion Jacobians. The results indicate high robustness and good visual quality.

**Strengths:**

- Image quality very high (impressive!)

- Method is novel and uses a lot of principled foundations (rooted in rigorous linear algebra concepts)

- Empirical results show promise (especially good for training-free)

**Weaknesses:**

- Seems like combination (all w/o rot.) and rotation are not easy. I would also be interested in the performance of All w/ rotation

- Baselines list is not comprehensive. Please include VINE [1] and other more recent SOTA watermarking methods. They tend to do better empirically.

- While I think your attack suite is good, I would recommend some additional ones from the WAVES suite (rinsing, embedding, etc.)


[1] https://arxiv.org/pdf/2410.18775

**Questions:**

1. I wonder if although robust is decent, watermark forgery could be a problem for this method. Could you run a small experiment on this?

2. Have you tried any other representation besides EOS (maybe some pooled representation)

---

### Official Review · Reviewer_sZdd · 2025-11-01

**Soundness:** 2
**Presentation:** 1
**Contribution:** 2
**Rating:** 2
**Confidence:** 4

**Summary:**

This paper proposes NullGuard, a diffusion-based watermarking framework that embeds imperceptible signals in the spectral null-space of pre-trained diffusion Jacobians. The goal is to mitigate semantic drift by performing latent-space embedding that minimally perturbs semantic content while enabling reliable detection.
The work is well-motivated and reports very strong empirical results, claiming improvements over existing method on both image quality and watermark detection rate (WDR).

**Strengths:**

* The motivation is clear and relevant: reducing semantic drift in diffusion watermarking via a principled latent-space embedding strategy.
* The method appears to achieve high image quanlity and high WDR simultaneously, outperforming prior methods across multiple attacks.

**Weaknesses:**

* **Method presentation is hard to follow.** The paper lacks a top-down overview of the full pipeline. Many modules are listed without clearly explaining each module’s role, or how modules connect logically within the overall framework. This makes the core design difficult to understand.
* **Missing or unclear definitions/abbreviations.** Several abbreviations are used without first being defined (e.g., SO(m), WDR).
* **Important terms are ambiguous or unjustified.**
  * *"zero-noise space"* (line 147): for diffusion models, noisy and noise-free latents are both in the same VAE latent space; introducing a separate "zero-noise space" is unnecessary and confusing.
  * *"semantic axes"* and *"insensitive axes"* (line 150): neither is defined precisely.
  * *"null-space"* itself is not formally defined in the paper: is it identical to the claimed "latent diffusion’s zero-noise space"? or derived from Jacobian near-null singular vectors? The text should make this explicit.
* **Figure/table quality and placement.** Figures are not vector graphics and become blurry when zoomed; Figure 2 is too small to read at normal scale. Tables/figures are placed far from their first references (e.g., Tables 1–2 appear on p.6 but are first refered on p.9), which disrupts readability.
* **Results beyond common sense without additional justification.** Table 2 reports very high PSNR (~54 dB) together with very high WDR across attacks, meanwhile the FPR is also very low in table 3. In this area, it is commonly observed that image quality and WDR usually trade-off with each other. In this area's common sense, an extremely small perturbation (to achieve very high PSNR) usually reduce robustness under strong attacks. If the watermark signal is encoded in a perturbation that is as small as this paper, then achieving high WDR under multiple attacks implies an unusually strong resistance to the pertubations. While one could inflate WDR by tolerating higher FPR, the paper claims low FPR as well. Without a clear information-theoretic or statistical explanation, these results are difficult to believe.

**Questions:**

1. Please provide a formal, self-contained definition of: (i) the **null-space** you use (with respect to which Jacobian and which operating point), (ii) the meaning of **“zero-noise space,”** and (iii) **“semantic axes”** vs **“insensitive axes.”** A block-diagram overview of the full pipeline (inputs/outputs per module) would also help.
2. Please define **SO(m)**, **WDR**, and other acronyms before using them.
3. How does your method avoid the usual **image-quality vs WDR** trade-off? Please provide a principled explanation.
4. Convert figures to vector graphics, enlarge small panels (e.g., Fig. 2), and place tables/figures close to their first mentions to improve readability.

---

### Note · Authors · 2025-11-13

I have read and agree with the venue's withdrawal policy on behalf of myself and my co-authors.